# CURIE: Evaluating LLMs On Multitask Scientific Long Context Understanding and Reasoning

**Hao Cui**[1,⋆] **Zahra Shamsi**[1,⋆] **Gowoon Cheon**[1,⋆] **Xuejian Ma**[1] **Shutong Li**[1] **Maria Tikhanovskaya**[2,◇]
**Peter Norgaard**[1] **Nayantara Mudur**[2] **Martyna Plomecka**[3,◇] **Paul Raccuglia**[1] **Yasaman Bahri**[1]
**Victor V. Albert**[4,5] **Pranesh Srinivasan**[1] **Haining Pan**[6] **Philippe Faist**[7] **Brian Rohr**[8]
**Michael J. Statt**[8] **Dan Morris**[1] **Drew Purves**[1] **Elise Kleeman**[1] **Ruth Alcantara**[1] **Matthew Abraham**[1]
**Muqthar Mohammad**[1] **Ean Phing VanLee**[1] **Chenfei Jiang**[1] **Elizabeth Dorfman**[1]
**Eun-Ah Kim**[9] **Michael P Brenner**[1,2] **Viren Jain**[1] **Sameera Ponda**[1] **Subhashini Venugopalan**[1,⋆,†]
[1]Google, [2]Harvard, [3]University of Zurich, [4]NIST, [5]UMD College Park, [6]Rutgers, [7]FU Berlin, [8]Modelyst, [9]Cornell

## Abstract

Scientific problem-solving involves synthesizing information while applying expert knowledge. We introduce CURIE, a scientific long-Context Understanding, Reasoning and Information Extraction benchmark to measure the potential of Large Language Models (LLMs) in scientific problem-solving and assisting scientists in realistic workflows. This benchmark introduces ten challenging tasks with a total of 580 problems and solution pairs curated by experts in six disciplines - materials science, condensed matter physics, quantum computing, geospatial analysis, biodiversity, and proteins - covering both experimental and theoretical workflows in science. We evaluate a range of closed and open LLMs on tasks in CURIE which requires domain expertise, comprehension of long in-context information, and multi-step reasoning. While Gemini Flash 2.0 and Claude-3 show consistent high comprehension across domains, the popular GPT-4o and command-R+ fail dramatically on protein sequencing tasks. With the best performance at 32% there is much room for improvement for all models. We hope that insights gained from CURIE can guide the future development of LLMs in sciences. Evaluation code and data links in: **https://github.com/google/curie**

## 1 Introduction

The advancement of science relies on the ability to build upon the collective knowledge accumulated in scientific literature, requiring not only deep domain expertise and reasoning skills, but also the capacity to apply that knowledge within the context of a given problem. Large Language Models (LLMs) have already shown remarkable knowledge across a wide spectrum of domains (Fig. 1b). Recent benchmarks e.g., MMLU (Hendrycks et al., 2020), have demonstrated proficiency in varied subjects (e.g., mathematics, history, social sciences, law) with standardized exams (BAR, SAT etc.), commonsense reasoning (Zellers et al., 2019), coding (Chen et al., 2021; Jimenez et al., 2023), reading comprehension (Dua et al., 2019), arithmetic (Zhang et al., 2024; Lu et al., 2023), and grade school scientific knowledge (Clark et al., 2018). However, as LLMs transition from merely surfacing knowledge to actively solving problems, the capacity to understand and reason about long-form, context-rich information is paramount. Recent advances in model architecture have seen dramatic increases in context windows from 8k to 32k, 128k, and 1M+ tokens, reflecting a growing recognition of this need.

This has led to development of benchmarks testing capabilities of LLMs on long document understanding such as ZeroScrolls (Shaham et al., 2023), Bamboo (Dong et al., 2023) on a variety of tasks including summarization (Kryściński et al., 2021), retrieval (Kamradt, 2023), multi-hop QA (Zhu et al., 2024; Kočiskỳ et al., 2018), sorting sequences (Wang et al., 2024) and others (Bai et al., 2023). However, current LLM benchmarks on science e.g., PubmedQA (Jin et al., 2019), GPQA (Rein et al., 2023) focus primarily on short sequence questions, with answers often in multiple choice form. To address this gap, we introduce the scientific long-Context Understanding, Reasoning, and Information Extraction benchmark (CURIE).

---

⋆equal technical contribution, ◇work done as a student researcher at Google
†Lead and corresponding author (vsubhashini@google.com)

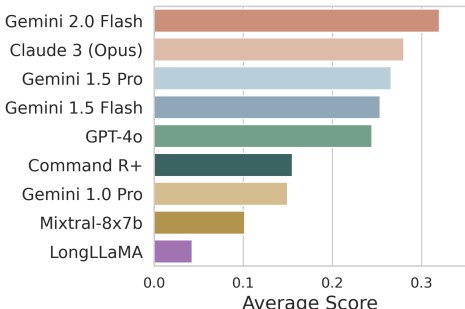 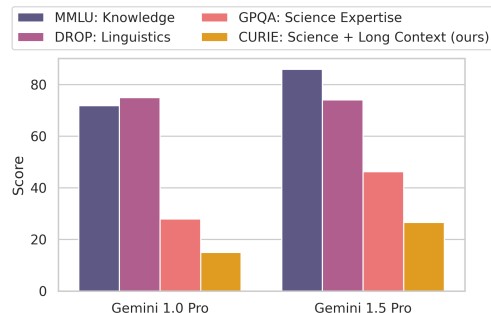

Figure 1: (a) Average normalized performance of state-of-the-art LLMs across 10 tasks from six scientific domains in CURIE. (b) Comparing performance of different model versions supporting long-context windows on previous benchmarks testing Knowledge (MMLU), Linguistic (DROP), and Science expertise (GPQA), along with our new scientific long-context understanding CURIE benchmark, highlighting the difficulty of the tasks in the benchmark.

The CURIE benchmark introduces 10 tasks, with a total of 580 input and solution pairs based on 429 research documents across six diverse scientific disciplines: materials science, theoretical condensed matter physics, quantum computing, geospatial analysis, biodiversity, and proteins – covering both experimental and theoretical aspects of scientific research. These tasks not only require deep domain understanding but also challenge models on their capacity to comprehend full-length scientific papers for information extraction, concept tracking, aggregation, algebraic manipulation, multimodal understanding, and cross-domain expertise (e.g. generating code for theoretical calculations).

We use the CURIE testbed to perform extensive evaluation and analysis of 8 state-of-the-art open and closed weight models (see Fig. 1a) supporting context windows of 32k tokens or more. Among closed models, Gemini Flash 2.0 and Claude-3 perform consistently well across all disciplines, while Command-R+ does better amongst the open models. Our materials tasks, which require models to exhaustively retrieve and aggregate information spread through the document proved to be exceptionally challenging for all models. Notably, while GPT-4o does well on most tasks, it fares dramatically poorly on the protein sequencing task failing to stop generation and repeating subsequences leading to a very low score. This repetition in subsequence is also seen in other open models. On the same task, the more recent Flash 2.0 provided correct python code to solve the problem half the time, and attempted to enumerate the sequence with lesser success other times.

While the CURIE benchmark is aimed at facilitating evaluation of scientific reasoning over long contexts, we hope the rich human annotations can serve the community in advancing planning, instruction following, and evaluation of generated texts of mixed and heterogeneous formats including dates, locations, numerical values, units, descriptors, domain specific terms, equations and code.

## 2    RELATED WORK

**Science NLP tasks.** There have been numerous datasets created to perform core NLP tasks on scientific texts e.g. BLURB Gu et al. (2021). This includes (i) *named entity recognition* to annotate entities such as disease names (Li et al., 2016) or material properties (Mavracic et al., 2021a), and relations such as disease-chemical interaction (Luan et al., 2018); (ii) *dependency parsing* (Kim et al., 2003), (iii) participant-intervention-outcome *(PICO) annotation* (Nye et al., 2018), (iv) *text classification* such as citation intent classification (Jurgens et al., 2018; Cohan et al., 2019), paper domain classification (Beltagy et al., 2019) from titles; (v) *relation extraction* for chemical-protein-disease annotation (Kringelum et al., 2016) or material structure and properties (Zhang et al., 2023), (vi) *information extraction* e.g. material property values (Dong & Cole, 2022a; Polak & Morgan), and (vii) *question answering* (Jin et al., 2019; Krallinger et al., 2020). However all of these focus on inputs of short length such as paper abstracts, sentences or text spans. They were curated for language models operating on short contexts, though the task in the actual scientific workflow requires application on full documents. Of the recent LLM benchmarks, GPQA (Rein et al., 2023) focuses on evaluating scientific domain expertise in biology, physics, and chemistry, while MMLU (Hendrycks et al., 2020) covers a range of high school science. These too are limited to short questions and multiple choice answers.

**Long context benchmarks.** With the increase in context windows supported by the LLMs, there have been new benchmarks focusing on evaluating long context capabilties. ZeroScrolls (Shaham

| Benchmark | Source Domain | Diverse Tasks | Long Context |
|---|---|---|---|
| ZeroSCROLLS | various | ✓ | ✓ |
| Bamboo | various | ✓ | ✓ |
| Ada-Level | lit., code | QA,sort | ✓ |
| Ruler | synthetic | ✓ | ✓ |
| BLURB | biomed | ✓ | ✗ |
| QASA | ML lit. | QA | ✓ |
| Qasper | NLP lit. | QA | ✓ |
| GPQA | science | MCQ | ✗ |
| CURIE | sci. lit. | ✓ | ✓ |

(a) Comparison with other datasets.

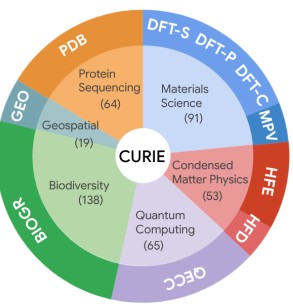

(b) Distribution of tasks.

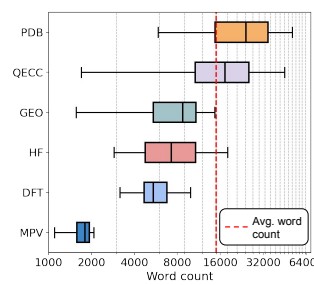

(c) Input word lengths per domain.

Figure 2: **CURIE dataset.** (a) CURIE introduces diverse long context tasks on scientific literature, (b) Distribution of 10 tasks and 429 research documents in six disciplines from which 580 examples were curated, and (c) Length of input context in each domain (log scale), avg. is about 15k.

et al., 2023) covers summarization, question answering, aggregation which are now present in many newer benchmarks: NIAH (Kamradt, 2023) includes retrieval, LongBench (Bai et al., 2023) includes bilingual tasks, Bamboo (Dong et al., 2023) has textual entailment tasks amongst others and M4LE (Kwan et al., 2023) has tasks testing translation and classification, L-eval (An et al., 2023) includes a task on multi-document dialog, and Loogle (Li et al., 2023) includes a computation task. However, while all of these benchmarks combine many existing datasets to cover a range of tasks none of them operate on data from the scientific domain. Further, for very long (100k+) contexts synthetically crafted data used, e.g. Ruler (Hsieh et al., 2024) proposes synthetically created length adaptable tasks and Ada-Level (Wang et al., 2024) includes length adaptable sorting and QA tasks, where they add distractor texts to increase the context. These ignore data from scientific literature that's naturally complex and requires processing long context, see Fig. 2(a) for comparison.

**Scientific literature.** Of the tasks most relevant to scientific expertise, QASA (Lee et al., 2023) and QASPER (Dasigi et al., 2021) operate on a full scientific paper, Machine Learning (ML) and Natural Language Processing (NLP) papers respectively, however these focus solely on question answering, since it is expensive and labor-intensive to collect tasks requiring expert knowledge. In our work we introduce ten new tasks curated from six disciplines, all annotated by experts(with Ph.D. degrees) and requiring reasoning over long context information, on average about 15k words (Fig. 2c). The outputs also include structured and unstructured text averaging about 1k words (Fig. 4c).

## 3 CURIE DATASET AND TASKS

The CURIE benchmark consists of a series of tasks that measure how well LLMs can assist in diverse scientific workflows, from synthesis of information towards final execution anchored on single scientific research documents. Each task in the benchmark: (1) is a realistic task performed by scientific experts on domains requiring years of study, (2) has information relevant to solve the given problem within the context provided (e.g. a full-length scientific paper, image/caption pair), and (3) ensures expert humans can evaluate task performance, providing metrics that highlight the potential limitations of current models. Summary of data, tasks, and metrics are in Appendix A.

**Collection guidelines.** We selected six domains requiring deep scientific expertise: materials science, theoretical condensed matter physics, quantum computing, geospatial analysis, biodiversity, and proteins. Within these, we worked with 1-3 experts to define tasks representative of realistic scientific workflows, covering the following seven assessment categories: *concept extraction, concept tracking (co-reference resolution), aggregation, algebraic manipulation, summarization, visual comprehension, and integrating expertise across domains.* We focused on tasks that, if successful, could enable automation (Donoho, 2024) of a time intensive critical component of a workflow e.g. extraction of experimentally reported values towards curating a database (Dong & Cole, 2022b), or generate code or calculations to fully reproduce computational or theoretical analyses. We worked with domain experts on 3 critical aspects of the task preparation: (1) sourcing papers representative of the task and domain; (2) creating ground truth answers that were accurate, nuanced and comprehensive; and (3) identifying measures to evaluate model responses against ground truth answers that properly captured salient features of the task which the experts also used for rating the difficulty of each example. Figure 2(b) shows the distribution and details of tasks in the CURIE benchmark.

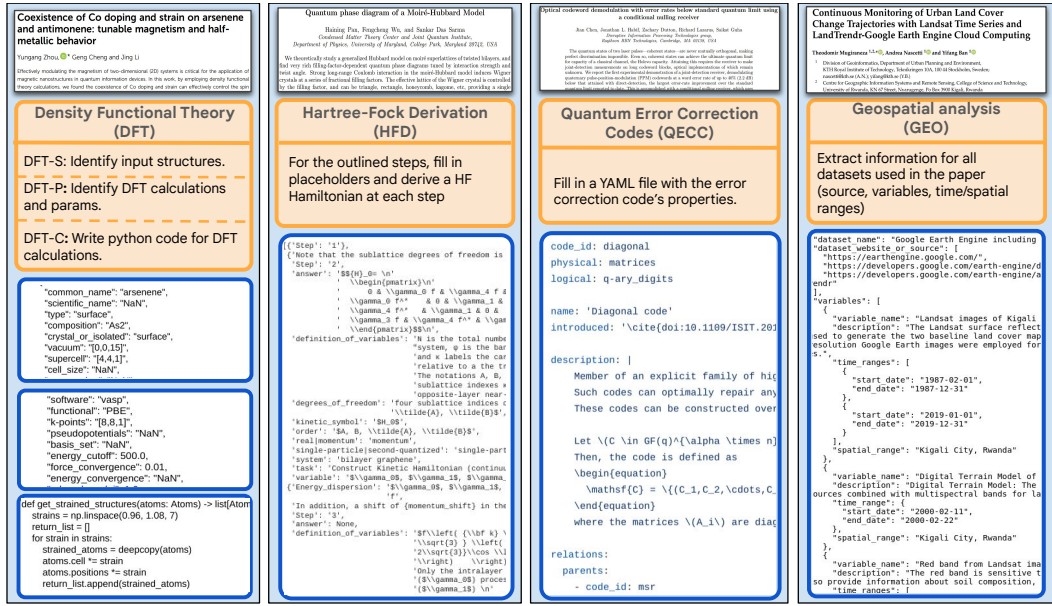

Figure 3: **Examples of tasks in the CURIE benchmark.** The DFT, HFD, QECC, and GEO tasks require the LLM to perform tasks on scientific papers (top blocks), as described in the task snippets (in orange), to extract, calculate, or aggregate information. Expected output (ground truth) snippets are shown in the blue blocks. (Only snippets of the query /outputs are shown for illustrative purposes.)

We provide a brief motivation and describe tasks for each of the domains. We identified papers with permissive licenses and worked with 1 to 3 experts in each domain to select specific papers, and to perform ground truth annotations. Data collection details are included in Appendix F.

**Density Functional Theory Task (DFT).** Density Functional Theory (DFT) is a widely used framework for quantum mechanical modeling of materials, enabling first-principles predictions and validation of experimental findings. We define 3 tasks that measure the ability of LLMs to reproduce DFT calculations from papers: (1) extracting input material structures (DFT-S); (2) identifying DFT calculation parameters associated with computation steps (DFT-P); and (3) translating computational steps essential for reproducing key results from the paper into functional code (DFT-C). Executing all these tasks successfully requires the LLM to comprehend domain-specific concepts, extract information dispersed across different sections of the publication, and generate scientific code. Two Ph.Ds with expertise in DFT computations identified 75 papers and prepared solutions for the tasks.

**Material Property Value Extraction (MPV).** While historically material properties have been tabulated in books (Welsch et al., 1993; Kutz, 2002), the published literature is an untapped resource, with experimentally reported materials, structure information, properties, and processing conditions. Human curation of such data is time intensive and expensive, and rule-based automation is limited in scope (Swain & Cole, 2016) as it can miss crucial processing details. However, prompt-based LLM extraction has shown promising early results (Zheng et al., 2023) at sentence-level extractions. Our benchmark contains 17 scientific papers for exhaustively extracting material properties at the full document-level, annotated and verified by experts. The main task is to identify all instances of material properties mentioned in the text, including material name, descriptor and particular property, along with the passage or table where the property is described. For ablations, we consider variations to extract just a pre-specified set of properties (e.g. just refractive index and band gap).

**Hartree-Fock Tasks (HFD, HFE).** In condensed matter physics, Hartree-Fock mean-field theory is a framework for simplifying mathematical descriptions of interacting quantum systems. The framework begins with an interacting Hamiltonian and uses symbolic computations to reduce to a simpler non-interacting Hamiltonian. We construct two tasks: derivation (HFD) and extraction (HFE). HFD measures the ability of an LLM to derive the Hartree-Fock mean-field Hamiltonian for a quantum many-body system, motivated by prior work (Pan et al., 2024). Deriving the correct answer requires 13-19 reasoning steps, making it extremely challenging without expert oversight as the LLM needs to (i) identify the necessary steps to carry out the calculation and (ii) execute the steps, which requires symbolic computations and deep knowledge of theoretical physics. The

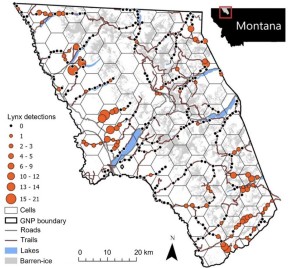 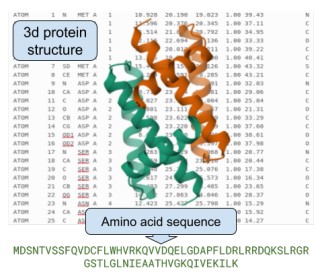 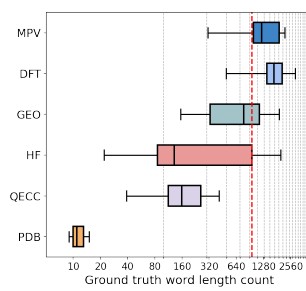

(a) An example from BIOGR.  (b) Illustration of the PDB task.  (c) Length of groundtruth outputs

Figure 4: (a) A map input from the BIOGR Task. (b) Example PDB task of reconstructing a protein's amino acid sequence from the 3D structure. (c) Length of ground truth per domain (avg. 954 words).

second simpler task, HFE, evaluates an LLM's ability to identify and aggregate key equations from a research paper to extract the most general mean-field Hamiltonian. We have 53 papers (38 HFE, 15 HFD) with expert annotations including prompts for detailed reasoning.

**Error Correction Zoo Task (QECC).** The Error Correction (EC) Zoo ᴱᶜ꜀꜀(Albert & Faist, 2024) is an open source effort to build a Wikipedia-like repository collecting and categorizing error correcting codes from the literature. EC codes are used to redundantly encode and store classical or quantum information so as to protect it from noise, with wide applications (e.g. in 5G communication and broadcast protocols). Creating an entry in the EC Zoo is a knowledge intensive process and requires listing the properties of a given EC code along with any relations to other codes in literature. Each example contains many different types of information: a succinct non-technical summary, bespoke technical details, numerical parameters quantifying code performance, and non-trivial connections to other literature. The diversity in encoding schemes makes the task of selecting critical components for code-entries challenging to templatize. We construct a benchmark that tests the ability of LLMs to curate the EC Zoo by taking a given paper and asking it to produce a YAML code-entry file. Our benchmark consists of codes from 65 papers curated by experienced experts.

**Geospatial Dataset Extraction (GEO).** Geospatial analysts integrate various diverse datasets to answer complex questions. For example, a study of time-series snowmelt detection over Antarctica may combine satellite imagery, radar data, weather station temperature data, elevation/topography information, etc (Liang et al., 2021). In this task, given a research paper, the LLM is required to identify all utilized datasets, including source websites, variable names, descriptions, time ranges and spatial ranges. This requires not only recognizing direct dataset references within the text, but also contextualization, comprehension and aggregation of information scattered throughout the paper, pushing the boundaries of long-context understanding. Our benchmark includes 19 papers ranging across earth observation, economics, epidemiology and public health, along with detailed ground truth annotations necessary to reproduce each study.

**Biodiversity Georeferencing Task (BIOGR).** Critical geospatial information is often conveyed exclusively through maps, highlighting the need for powerful tools capable of interpreting such visual data. In this task we study the ability of multimodal LLMs to work with geographical maps. Specifically, we investigate the core capability of georeferencing, where, given an image of a map and its associated caption, the task is to determine the latitude/longitude bounding box encompassing the region displayed. A domain expert would often use a multi-step process and specialized mapping tools (e.g., QGIS, ArcGIS), zooming in and switching between different imagery layers to find recognizable landmarks (river bends etc.) For this multimodal task, we assembled a dataset of 138 map images (e.g., Fig. 4a) and captions from papers in ecology, of varying difficulty with ground truth labels for bounding boxes verified by domain experts.

**Protein Sequence Reconstruction (PDB).** This final task tests the ability of an LLM to extract meaning from a three dimensional structure, associating the 3D structure of a protein with its sequence. Given the 3D structural coordinates of a protein, provided in the Protein Data Bank (PDB), capturing the precise arrangement of atoms within a complex molecule, we ask the LLM to reconstruct the protein's amino acid sequence. To create the benchmark we curated the DB file to contain only coordinate information, stripped of any explicit functional annotation. This minimalist approach forces the LLM to rely solely on its understanding of structural patterns to deduce the underlying amino acid sequence. We curated 64 structures with annotations for this task.

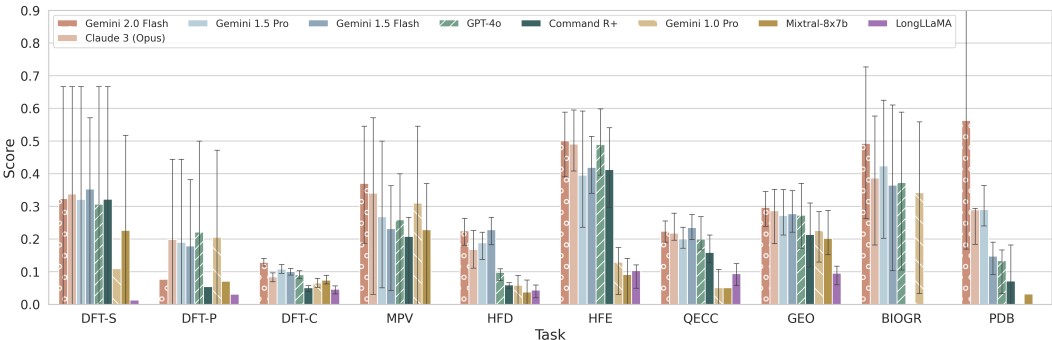

Figure 5: **Per task normalized scores** of various LLMs on the CURIE benchmark that measures performance of LLMs on 10 long-context tasks requiring expertise across six scientific disciplines. DFT-S, DFT-P, and MPV are scored using LLMSim, while others use programmatic metrics.

**Annotations.** The annotations include ground truth solutions for each task, and difficulty rating for each example based on complexity of information extraction and reasoning (details in Appendix F). On the extraction tasks, such as MPV, HFE, GEO, and DFT-S, experts within each domain reviewed each others' work and reported a high rate of agreement.

## 4 EXPERIMENTS AND EVALUATION

### 4.1 EXPERIMENTAL SETUP

We evaluate the CURIE benchmark tasks on several state-of-the-art LLMs supporting long-context windows, including five closed weight LLMs such as GPT-4o (OpenAI), Claude-3 Opus (Anthropic), Gemini 1.5 Pro (Reid et al., 2024), and three open-weight LLMs, Mixtral (Jiang et al., 2024), Command-R-+ (Cohere), and LongLLaMa-3B (Tworkowski et al., 2024). We follow a standard zeroshot prompt template across tasks, first describing the task the model needs to perform and the desired output format, and then providing the text of the full paper (except for LongLLaMa-3B full paper was provided first). In the case of DFT and MPV tasks, we provide the output format in the context of an additional hand-crafted excerpt to clarify expectation of formats for each field. The BIOGR task is multimodal, and for this we provide just the image and caption as input, rather than the full paper. Performance is reported for each model on each task using a single run.

### 4.2 EVALUATION METRICS

For the tasks requiring generation of long English text responses (eight out of ten tasks), we use the ROUGE-L (Lin, 2004) and BERTScore F1 (Zhang et al., 2019) metrics. For BIOGR and PDB we use other more appropriate quantitative metrics. BIOGR uses Intersection-over-Union (IoU), defined as the area of intersection between two bounding box regions (using latitude and longitude) divided by the union. This metric captures the similarity between two polygons, accounting for both location and size, while also being scale-invariant. We also present additional metrics such as the normalized distance error between the predicted and ground truth boxes in Appendix Tab. 7. For the PDB task, we compare reconstructed sequences to ground truth sequences (Cock et al., 2009), using pairwise sequence alignment scored using the number of identities. The raw scores are normalized by the alignment length to account for potential length discrepancies, yielding the identity ratio $(ID_r)$ metric.

**Computing average performance across all tasks in the CURIE benchmark.** Appendix Table 4 reports the primary metric for each task used to compute the average performance. DFT-S, DFT-P, MPV use model-based F1 scores (Sec. 4.3), BIOGR uses IoU, PDB uses $ID_r$, the rest use ROUGE-L. All metrics are normalized to [0,1]. Fig. 5 shows performance of models per-task and Fig. 1(a) shows average performance of models across all 10 tasks.

**Human evaluations.** Additionally, experts identified a set of rubrics for evaluating model responses and compared responses. E.g, for the tasks requiring extraction of information such as DFT-S, DFT-P, and MPV, experts measured precision and recall of retrieved information relative to ground truth. Experts also rated 10-12% responses in each task with an overall "good", "ok", "bad" rating.

## 4.3 MODEL-BASED EVALUATION OF MIXED LONG FORM RESPONSES

Tasks in CURIE are varied and have ground truth annotations in mixed and heterogenous outputs. Evaluating free-form generation is challenging because answers are often descriptive, and even when a format is specified as in most of our cases the response to each field can have differing forms e.g. in case of materials grid points may sometimes be specified as [p,q,r] and at other times as p×q×r. Hence most existing knowledge related benchmarks (Hendrycks et al., 2020; Rein et al., 2023) lean towards multiple choice format for answers. While these allow for clean evaluation, this doesn't allow us to evaluate the full expressiveness of the model. Inspired by the ability of LLMs to evaluate natural language (Zheng et al., 2024; Liu et al., 2024; Kamalloo et al., 2023), three recent approaches, LAVE (Mañas et al., 2024), LIMA (Zhou et al., 2024) and Prometheus-Vision (Lee et al., 2024), utilize the in-context capability of instruction-tuned LLMs to rate the candidate answers in 3-point, 6-point and 5-point Likert scales, respectively. While evaluation on Likert scales is suited for generated text, many of the outputs on our task have structured information that could benefit from more fine grained evaluation. So we propose two model-based evaluations $(i)$ LLMSim a nuanced score for measuring similarity of elements in lists of dictionaries, which can then be used to compute precision and recall of elements, and $(ii)$ LMScore an overall weighted score on a 3-point scale obtained by asking the LLM if the predictions match ground truth. We focus on LLMSim here since it provides a quantitative measure for structured outputs missing in previous work and discuss LMScore in Appendix D.

**LLMSim** is used to compare similarity of dictionary elements to assist in comparison of sets of dictionaries. Our goal is to identify the number of ground truth dictionary items that have been retrieved correctly. So, we ask the LLM to examine all of the predicted dictionaries and match and identify the predicted dictionary most similar to the each of the ground truth dictionaries. We use a chain-of-thought (CoT) prompt that asks the LLM to identify the predicted dictionary indices that correctly match each field (key) of the ground truth, and then select the predicted dictionary index most similar to the ground truth or output 'None'. Prompt and code are included in the supplement.

Concretely, suppose $D_P$ is the set of predicted dictionaries, $D_G$ the set of ground truth dictionaries, LLMSim helps find the optimal matching $M$ between the predicted dictionaries and each ground truth $D_g \in D_G$:

$$\text{LLMSim} = M(D_P, D_g)$$

$$= \begin{cases} None, \text{if no match in values} \\ D_p \in D_P : \arg\max \ s(f_i, D_p, D_g) \end{cases}$$

where $f_i$ represents the $i^{th}$ field (key) in the dictionary and $s(f_i, D_p, D_g)$ is the similarity of the value of each field of $D_p$ with $D_g$. Given the matching, we can then compute precision, recall and F1 as

$$Pr = \frac{|(D_p, D_g) \in M|}{|D_P|}, Re = \frac{|(D_p, D_g) \in M|}{|D_G|}, F1 = 2\frac{Pr \cdot Re}{Pr + Re}$$

## 5 RESULTS

**Main Results.** Fig. 1 shows the performance of all models averaged across all tasks in the CURIE benchmark. Claude-3 Opus is the best performing with consistent high performance across all tasks. Fig. 5 and Table 1 show task level performance of all models. The popular GPT-4o outperforms the others on GEO and BIOGR, however it's performance on PDB and HFD is surprisingly low. On closer inspection we found the GPT-4o model exhibited repetition in the outputs in the PDB task (see. Fig. 6), clipped responses in the MPV task, and failed to follow formatting instructions on the HFD task leading to lower performance. Overall though, most of the closed models had similar performance, and given the variability (e.g., 25%-75% error bars around the mean in Fig. 5) the difference between them is not significant. On several tasks there is considerable room for improvement, making CURIE an interesting benchmark for furthering model development.

**Room for improvement.** Fig.1(b) compares performance of models from two different generations, Gemini 1.0 pro (32k) and Gemini 1.5 pro (1M+ context window) on popular benchmarks evaluating linguistic capability Dua et al. (2019), breadth of knowledge (Hendrycks et al., 2020), and expertise in science (Rein et al., 2023), alongside the performance on our benchmark evaluating expertise with long-context comprehension. We observe that there is considerable room for improvement on the types of realistic complex scientific tasks the CURIE benchmark provides.

| Method | DFT | | MPV | | HFD | | HFE | | QECC | | GEO | | BIOGR | PDB |
|---|---|---|---|---|---|---|---|---|---|---|---|---|---|---|
| | R-L | B-F1 | R-L | B-F1 | R-L | B-F1 | R-L | B-F1 | R-L | B-F1 | R-L | B-F1 | IoU | $ID_r$ |
| *Zero-shot Open Weight LLMs* | | | | | | | | | | | | | | |
| Mixtral | 12.20 | 0.75 | 12.48 | 0.82 | 3.78 | 0.27 | 9.15 | 0.47 | 5.11 | 0.17 | 20.23 | 0.67 | 0.00 | 0.03 |
| Command-R+ | 13.79 | 0.79 | 15.67 | 0.83 | 5.93 | 0.82 | 41.23 | 0.85 | 15.88 | 0.67 | 21.36 | 0.78 | 0.00 | 0.07 |
| LongLLaMa | 8.17 | 0.77 | 9.28 | 0.80 | 4.36 | 0.78 | 10.33 | 0.77 | 9.38 | 0.76 | 9.53 | 0.77 | 0.00 | 0.00 |
| *Zero-shot Closed Weight LLMs* | | | | | | | | | | | | | | |
| Gemini 1.0 Pro | 11.22 | 0.78 | 17.37 | 0.83 | 5.81 | 0.53 | 12.95 | 0.69 | 5.10 | 0.27 | 22.56 | 0.79 | 0.34 | 0.00 |
| GPT-4o | 13.30 | 0.78 | 12.74 | 0.83 | 9.81 | 0.81 | 48.93 | 0.85 | 20.02 | 0.70 | 27.30 | 0.80 | 0.37 | 0.13 |
| Gemini 1.5 Pro | 13.66 | 0.78 | 31.54 | 0.82 | 18.86 | 0.84 | 39.56 | 0.71 | 20.07 | 0.76 | 27.24 | 0.74 | 0.42 | 0.29 |
| Gemini 1.5 Flash | 11.31 | 0.76 | 12.11 | 0.79 | 22.86 | 0.84 | 41.92 | 0.78 | 23.50 | 0.81 | 27.76 | 0.78 | 0.36 | 0.15 |
| Gemini 2.0 Flash | 14.18 | 0.79 | 14.72 | 0.84 | 22.52 | 0.84 | 50.06 | 0.87 | 22.36 | 0.79 | 29.69 | 0.79 | 0.49 | 0.56 |
| Claude 3 (Opus) | 13.78 | 0.80 | 15.86 | 0.83 | 16.82 | 0.83 | 49.10 | 0.87 | 21.81 | 0.72 | 28.66 | 0.79 | 0.39 | 0.29 |

Table 1: **Results comparing performance of all models on all tasks based on automated metrics**
R-L: Rouge-L, and B-F1:BertScore-F1. The avg. performance of all 3 DFT tasks are reported under
DFT. All models support a context length of 32k or more. BIOGR has multimodal inputs which is
unsupported by the chosen open models. Blue highlights the highest values.

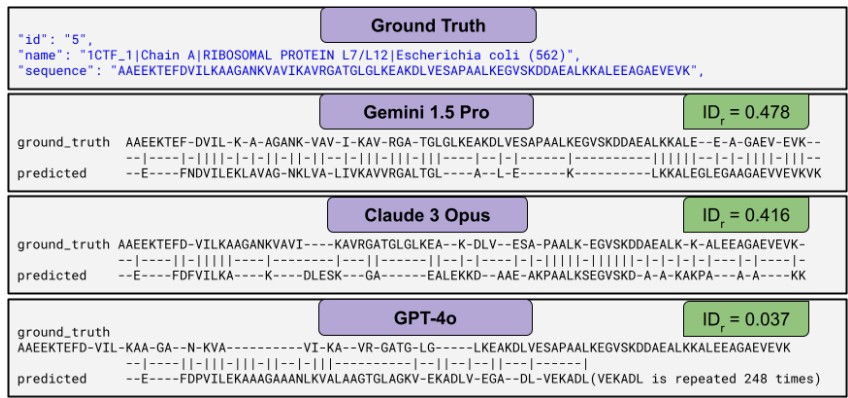

Figure 6: **Comparing responses on the PDB task**. Measuring alignment using identity ratio ($ID_r$)
between the predicted amino acid sequence and groundtruth sequence for a given protein structure,
Gemini 1.5 pro and Claude were better at predicting the sequence of amino acids, whereas GPT-4o
collapsed into a mode of repetition, and Flash 2.0 generated code to solve the problem.

**Human vs Model-based eval on retrieval.** On the information retrieval tasks: DFT-S and DFT-P
tasks which requires LLMs to retrieve material structures and DFT parameters from a given paper;
as well as the MPV tasks requiring models to retrieve material property and values, we use LLMSim
to compare the dictionaries of extracted material properties. Table 2 reports precision, recall and F1
scores computed after matching elements using LLMSim. We found the precision and recall to
closely match those measured by human experts (on the Gemini 1.5 pro and GPT-4o models).

**Human evaluation of model responses.** We worked with experts in each domain to evaluate pre-
dictions generated by the models against ground truth responses on a 3-point scale identical to the
proposed LMScore (in Appendix D). For each example, the expert was asked to rate a response as
"good" if it had few or no errors compared to the ground truth, "okay" if it had many minor errors,
and "bad" if there were major errors. We use these human responses to compare and correlate the
newly proposed LMScore which is reported in the Appendix (App. Fig. 10 and Table 5). While
LMScore appears to be promising, it requires further analysis prior to wider usage.

**Performance vs. Difficulty** Experts in each domain independently determined and rated the diffi-
culty of answering each input example on a 3-point scale as "easy", "medium", or "hard". In most
cases, such as with the MPV, HFE, QECC, and DFT-S extraction and aggregation tasks, difficulty
was determined based on how dispersed the information was within the paper. "Easy" examples
had answers often within a section or a page, while "medium" cases could be spread across mul-
tiple sections, and if the information required knowledge of specific literature outside of the given
context the example was rated "hard". Additionally for the DFT-C code generation task, the ratio of
the number of implementable functions to the total number of functions mentioned in the paper was
used. For HFD, the number of reasoning steps and for GEO the number of datasets was also fac-

tored in, and for PDB the length of the sequence determined complexity. Fig. 7 reports performance of each model sliced by difficulty. Overall, models perform substantially better on easy examples compared to the medium and hard examples. Models appear to perform about the same on examples marked medium or hard. Though, one thing of note is that there are usually many more medium examples than hard examples across all tasks.

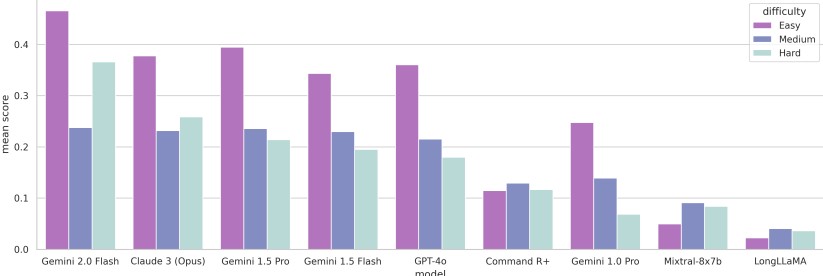

Figure 7: **Avg. performance of models sliced by difficulty of examples.** Consistent with expectations, all models perform substantially better on easy examples except in the case of Mixtral. For most domains, experts independently converged on measuring difficulty for each example based on how spread-out the requested information was within the context of the full paper.

## 6 DISCUSSION

**Model responses lack robustness in instruction following.** A common observation across tasks was that, there is variability across model runs, even though average performance remained fairly constant. The variability is usually higher on harder tasks. Instruction following remains a challenge: Models often had *pieces* of the right answer, but were unable to consistently format it despite examples in the prompt. In rare cases, even though we explicitly asked models to provide answers when they weren't sure, they often refused to venture an educated guess. The model-based evaluation metrics were quite helpful in mitigating issues arising from lack of adherence to instructions.

**Performance on retrieval.** Fig. 5 and Table 1 report performance of the models on each task in each of the domains. Noticeably all models show high ROUGE-L scores on HFE which is a variation of the needle-in-a-haystack problem where the model needs to extract related equations that might be spread throughout the paper. On tasks requiring exhaustive retrieval of multiple values and aggregation, e.g., DFT (see App. Fig. 14), MPV, and GEO, the models have considerably lower performance than just single value retrieval tasks (e.g., HFE).

**Concept tracking, aggregation, summarization.** On tasks requiring concept aggregation and tracking, e.g. DFT-P, GEO and QECC, experts found responses from some models quite promising. With DFT-P, Claude-3 appeared to understand the purpose of DFT calculations better and grouped relevant parameters to appropriate functions (see App. Fig.8). On QECC, the experts noted that the summaries generated by the LLM tended to be succinct while also including multitude of key informational "nuggets" and quantitative measurements. While not all of these were correct or important, experts noted that it would be easier to exclude the wrong bits (after examination) but harder to extract and comb out such details from the paper. On the GEO task, the closed models did well

| Model | DFT-S | | | DFT-P | | | MPV | | | MPV-non-trivial | | | MPV-specific | | |
|---|---|---|---|---|---|---|---|---|---|---|---|---|---|---|---|
| | Pr. | Rec. | F1 | Pr. | Rec. | F1 | Pr. | Rec. | F1 | Pr. | Rec. | F1 | Pr. | Rec. | F1 |
| *Zero-shot Open Weight LLMs* | | | | | | | | | | | | | | | |
| Mixtral | 24.96 | 23.30 | 22.67 | 9.12 | 6.13 | 7.09 | 31.86 | 23.29 | 22.82 | 29.70 | 21.14 | 22.31 | 22.20 | 35.05 | 22.64 |
| Command-R+ | 41.67 | 27.95 | 32.19 | 6.92 | 4.63 | 5.41 | 22.64 | 27.25 | 20.80 | 3.87 | 6.31 | 4.52 | 18.18 | 17.84 | 15.97 |
| LongLLaMa | 1.26 | 1.47 | 1.36 | 2.99 | 3.95 | 3.13 | 0.00 | 0.00 | 0.00 | 0.00 | 0.00 | 0.00 | 0.00 | 0.00 | 0.00 |
| *Zero-shot Closed Weight LLMs* | | | | | | | | | | | | | | | |
| Gemini 1.0 Pro | 11.19 | 12.62 | 10.93 | 23.1 | 21.01 | 20.56 | 31.28 | 32.92 | 31.00 | 36.41 | 34.92 | 31.78 | 24.86 | 38.76 | 23.26 |
| GPT-4o | 36.96 | 29.50 | 30.63 | 27.93 | 19.66 | 22.13 | 39.22 | 24.14 | 25.90 | 45.10 | 24.08 | 30.05 | 32.35 | 21.77 | 22.97 |
| Gemini 1.5 Pro | 36.04 | 33.67 | 32.11 | 23.67 | 16.53 | 19.00 | 23.86 | 38.36 | 26.85 | 31.74 | 42.60 | 30.08 | 25.00 | 31.34 | 24.48 |
| Gemini 1.5 Flash | 33.07 | 48.74 | 35.28 | 22.35 | 16.42 | 17.91 | 16.41 | 50.90 | 23.16 | 15.82 | 50.97 | 21.69 | 14.77 | 32.90 | 17.76 |
| Gemini 2.0 Flash | 31.38 | 40.46 | 32.39 | 8.22 | 7.74 | 7.68 | 35.84 | 46.56 | 36.99 | 30.81 | 47.76 | 33.79 | 26.48 | 33.64 | 24.37 |
| Claude 3 (Opus) | 40.45 | 32.89 | 33.76 | 27.26 | 17.17 | 19.87 | 41.35 | 35.60 | 34.04 | 45.64 | 43.67 | 38.32 | 32.18 | 47.06 | 31.48 |

Table 2: **Retrieval performance using LLMSim** On tasks requiring exhaustive retrieval of information we use LLMSim and compute Precision, Recall, and F1 scores on each document and report the mean. We also include 2 ablations for the MPV task where we ask the LLM to retrieve non-trivial or specific property values (refractive index and optical bandgap) for materials.

Figure 8: **Example of model outputs for the DFT-P** parameter identification task. Claude-3 Opus appears to understand the purpose of the calculations better than the other models and avoids unnecessary repetition. Claude-3 correctly (green) identifies that there is one set of DFT parameters used in the actual study as well as two more set of parameters which are used for convergence testing.

to extract some of the important datasets with the correct spatial and temporal ranges (Fig.25) but performance degrades when multiple datasets are used to cover a larger spatial extent (App. Fig.27). Overall, carefully engineered prompts and agentic workflows could be effective on such tasks.

**Closed vs Open models.** One thing of note is that on QECC, DFT, and MPV extraction and aggregation tasks, the Command-R+ open weights model which uses retrieval-augmented approaches shows performance similar to the closed weight models. The evals on BIOGR highlight that open models are yet to support both multimodal and long-context capabilities which can enable more scientific applications. On PDB, Mixtral performed higher than GPT-4o which is quite surprising. Both LongLLaMA and Command-R+, failed to produce any sort of FASTA format on the PDB task. They either failed to fully understand the task, or missed steps during aggregation.

**Reasoning in code.** On the PDB task, remarkably, the more recent Gemini Flash 2.0 model, not only understood the task but seemed to realize the task is better solved by using programming and wrote a python function (which executed correctly) about half of the time. On other half of the PDB inputs, Flash 2.0 decided to aggregate and generate the amino acid sequence similar to other models.

Overall, across tasks, model performances have room for improvement and we discuss specific examples and failure cases in appendix F for each task.

## 7 CONCLUSION

In this work we introduce the CURIE benchmark. A series of tasks designed to measure the ability of LLMs on scientific reasoning and understanding long-contexts. Our main contributions are $(i)$ A new benchmark of 580 examples from 429 research papers that can assess LLMs on comprehension of long-context information from across six scientific disciplines requiring deep expertise. $(ii)$ 10 realistic tasks combining concept retrieval and extraction, concept tracking, aggregation, algebraic manipulation, and expertise across multiple domains to measure capability of models on different aspects of scientific workflows. $(iii)$ We propose model-based evaluation metrics to address the challenge of automatically evaluating complex mixed-format heterogeneous outputs and compare them with human evaluations. $(iv)$ We share guidelines for curating such multi-step tasks and evaluating annotation quality of such complex answers. We hope the diverse tasks and rich annotations in the CURIE benchmark can serve the community in not only evaluating LLMs on their scientific problem solving abilities but also advance research on scientific planning, instruction following, and evaluation of generated texts containing information of diverse types and formats.

## REPRODUCIBILITY STATEMENT

We make the data, prompts, and code available in https://github.com/google/curie under the Apache 2.0 license. Our dataset is available under a CC-BY license. The dataset includes full text of the papers, the ground truth annotations, prompts (which includes prompts used to elicit model responses, as well as the prompts used for evaluation using the proposed LMScore and LLMSim metrics), model responses and code for evaluations. The appendix includes additional information on each of the tasks, annotation procedures, and examples of model outputs and failure modes.

## DISCLAIMER

V.V.A. participated only as a subject-matter expert for the QECC task. Certain equipment, instruments, software, or materials are identified in this paper in order to specify the experimental procedure adequately. Such identification is not intended to imply recommendation or endorsement of any product or service by NIST, nor is it intended to imply that the materials or equipment identified are necessarily the best available for the purpose.

## ACKNOWLEDGEMENTS

N.M. acknowledges support from the National Science Foundation under Cooperative Agreement PHY2019786 (The NSF AI Institute for Artificial Intelligence and Fundamental Interactions).

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

# A    DATASET STATISTICS AND TASKS OVERVIEW.

In Fig. 2 (b), Fig. 2 (c), and Fig. 4 (c) we present the distribution of the CURIE dataset in terms of papers, and distribution of lengths of inputs and outputs per domain. Here, in Table. 3 we consolidate and present the number of examples in each task and the average length of the inputs and ground truth outputs. Table 4 presents a brief description of the tasks, the capabilities necessary for the task, the format of the output, and both programmatic and LLM-based metrics that are used to evaluate performance on the task.

| Domain | Task | # papers | # examples | Avg. length (# words) | |
|---|---|---|---|---|---|
| | | | | input | output (ground truth) |
| Material Science | DFT-S | 74 | 74 | 5818 | 232 |
| Material Science | DFT-P | 74 | 74 | 5818 | 132 |
| Material Science | DFT-C | 74 | 74 | 5818 | 1742 |
| Material Science | MPV | 17 | 17 | 1687 | 2188 |
| Condensed Matter Physics | HFD | 15 | 15 | 5385 | 1422 |
| Condensed Matter Physics | HFE | 38 | 38 | 8472 | 111 |
| Quantum Computing | QECC | 65 | 65 | 19913 | 207 |
| Geospatial | GEO | 19 | 19 | 7802 | 808 |
| Biodiversity | BIOGR | 137 | 138 | - | 20 |
| Protein Sequencing | PDB | 64 | 64 | 44028 | - |

Table 3: **Statistics of the CURIE dataset.** We report the the number of papers and examples for each task in each of the domains and also include the average length the input and ground truth outputs in words.

| Task | Brief description | Capability | Output Format | Primary Eval. Metric |
|---|---|---|---|---|
| DFT-S | Extract input material structures for DFT calculations. | entity recognition, concept tracking | JSON | LLMSim-F1 |
| DFT-P | Extract parameters for DFT calculations. | concept extraction, tracking, aggregation | JSON | LLMSim-F1 |
| DFT-C | Write functional code for DFT computations. | concept aggregation, coding | TEXT | ROUGE-L |
| MPV | Identify all instances of materials,their properties, and descriptors. | entity recognition, concept extraction, tracking | JSON | LLMSim-F1 |
| HFD | Derive the Hartree-Fock mean-field Hamiltonian for a quantum many-body system. | concept extraction, alg. manipulation, reasoning | TEXT | ROUGE-L |
| HFE | Extract the most general mean-field Hamiltonian. | concept extraction | TEXT (latex equation) | ROUGE-L |
| QECC | Create a YAML file with the Error Correction Code's properties. | concept aggregation, summarization | YAML | ROUGE-L |
| GEO | Extract information for all geospatial datasets used along with the spatial and temporal extents. | concept extraction, aggregation | JSON | ROUGE-L |
| BIOGR | Determine the latitude, longitude bounding box encompassing the region in the map image. | visual comprehension, reasoning | JSON (lat., lon. co-ordinates) | Intersection-over-union (IoU) |
| PDB | Reconstruct a protein's amino acid sequence form the 3D structure. | tracking, aggregation reasoning | Code or TEXT (seq.) | Identity ratio $(ID_r)$ |

Table 4: **Task details and capabilities required.** A brief description of the tasks, capabilities assessed, output format, and the primary evaluation metric (used in Fig. 1 and Fig. 5).

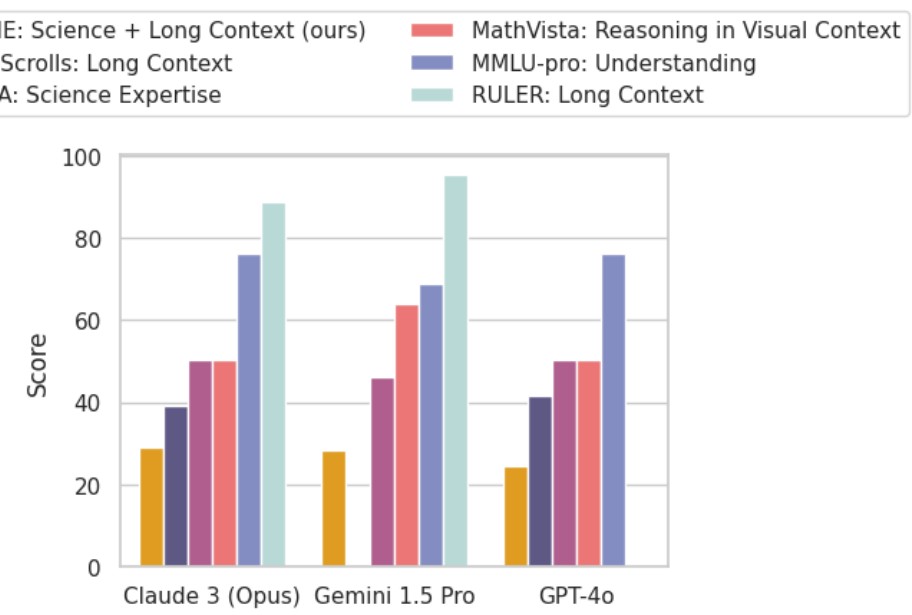

Figure 9: **Comparing model performances on scientific QA and long-context benchmarks.** We compare the performance of the top 3 closed models on CURIE against, 2 science QA benchmarks - GPQA, MathVista, an improved language understanding benchmark MMLU-pro, and 2 long-context benchmarks Zero-Scrolls and RULER.

## B  MODEL PERFORMANCES ON SCIENCE VS LONG-CONTEXT BENCHMARKS

In Fig. 9 we observe how performance of models on the CURIE benchmark correlate or compare with performance on other scientific QA benchmarks and long-context benchmarks. We specifically examine the top-3 models on CURIE (Claude-3, Gemini 1.5 Pro, and GPT-4o). We observe that performance of models on CURIE is more in-line with the realistic ZeroScrolls long-context benchmark and also the GPQA scientific question answering benchmark. CURIE appears to be a harder benchmark than both Zero-Scrolls and GPQA.

## C  ANNOTATOR AGREEMENTS.

For all tasks we worked with experts to establish a consistent output format over a couple of initial iterations. We then had a pilot phase followed by some adjudication to clarify outputs. To evaluate annotator agreements, for retrieval tasks, we had annotators examine each other's work after the adjudication. For the HFE and DFT-S tasks the agreement was near perfect ($> 95\%$). There were minor differences in the exact phrasing / chemical formula notation used. For MPV, we had two rounds of feedback and were then able to get an agreement over 80% on the materials to be extracted comprehensively. For GEO, the agreement was again high after an initial pilot phase and discussion, agreement was over 90% on the datasets, spatial and time ranges. There were minor differences in the exact phrasing of the responses but these were discounted e.g. "for each year between 2012-2015" vs "2012,2013,2014,2015".

## D  LMSCORE: A COARSE MODEL-BASED EVALUATION METRIC.

Evaluation on Likert scales provide a quick coarse signal of the quality of the responses on generation tasks. With LMScore we propose an overall weighted score on a 3-point scale obtained by asking the LLM if the predictions match ground truth and using the model's confidence (log-probs scores) to get a weighted score. Specifically, given the ground truth and predicted responses, we ask the model to check if the predicted responses match the ground truth, and ask the model to output "good" (if the prediction has few minor errors), "okay" (if there are many minor errors), and "bad" if

there are major errors. Instead of using the model generated response directly, we compute a score based on the model log-likelihood values. If $x_t$ represents the tokens for the 3 categories we are interested in, $x_t \in \{bad, ok, good\}$, and $w_t$ are the corresponding weights we want to assign to each category, $w_t \in \{0, 0.5, 1\}$, then

$$LMScore = \sum_{t=0}^{2} p(x_t) \times w_t \tag{1}$$

$p(x_t)$ is computed by renormalize the probabilities of the tokens by considering a $softmax()$ operation on the log-probabilities of the tokens: $([l_{bad}, l_{ok}, l_{good}])$. We consider an uncased version of the tokens and treat 'ok' and 'okay' equivalently. We use GPT-4o as the LLM. When the tokens are not present in the top-5 log probabilities we compute an approximation based on the probability mass of the tokens not present in the top-5. We make the code for computation available in the supplement.

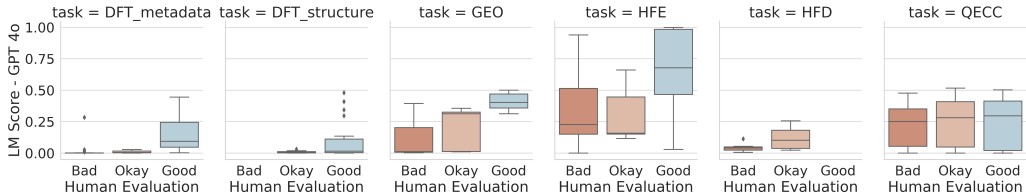

Figure 10: **Correlation of GPT-4o based LMScore metric with human evaluations,** Across tasks in domains where ROUGE-L is the primary evaluation metric, LMScore appears to be a promising alternative to ROUGE.

| Method | DFT | | MPV | | HFD | | HFE | | QECC | | GEO | |
|---|---|---|---|---|---|---|---|---|---|---|---|---|
| | R-L | LMS | R-L | LMS | R-L | LMS | R-L | LMS | R-L | LMS | R-L | LMS |
| *Zero-shot Open Weight LLMs* | | | | | | | | | | | | |
| Mixtral | 7.43 | 0.08 | 13.43 | 0.18 | 3.78 | 0.05 | 9.15 | 0.11 | 3.10 | 0.06 | 20.23 | 0.19 |
| Command-R+ | 5.12 | 0.06 | 11.22 | 0.37 | 15.81 | 0.17 | 41.23 | 0.31 | 17.23 | 0.16 | 21.36 | 0.15 |
| LongLLaMa | 4.61 | 0.02 | 6.03 | 0.06 | 4.36 | 0.01 | 10.33 | 0.00 | 8.67 | 0.01 | 9.53 | 0.00 |
| *Zero-shot Closed-Weight LLMs* | | | | | | | | | | | | |
| Gemini 1.0 Pro | 10.52 | 0.06 | 13.69 | 0.30 | 18.10 | 0.02 | 36.60 | 0.10 | 17.48 | 0.05 | 28.44 | 0.18 |
| GPT-4o | 9.05 | 0.12 | 13.52 | 0.25 | 9.81 | 0.06 | 48.93 | 0.47 | 19.51 | 0.16 | 31.47 | 0.30 |
| Claude 3 (Opus) | 8.38 | 0.11 | 15.54 | 0.34 | 16.82 | 0.13 | 49.10 | 0.50 | 20.54 | 0.18 | 28.63 | 0.24 |
| Gemini 1.5 Flash | 10.62 | 0.13 | 14.14 | 0.43 | 17.53 | 0.21 | 43.94 | 0.39 | 17.40 | 0.22 | 28.94 | 0.29 |
| Gemini 1.5 Pro | 10.83 | 0.12 | 14.12 | 0.41 | 18.86 | 0.14 | 39.56 | 0.33 | 18.03 | 0.23 | 30.25 | 0.30 |

Table 5: Results comparing performance of all models on all tasks based on the ROUGE-L metric and the proposed GPT-4o based LMScore (LMS) metric. DFT reports avg. across all 3 DFT tasks.

**Human vs. Model-based 3-point evaluations** We worked with experts in each domain to evaluate predictions generated by the models against ground truth responses on a 3-point scale identical to the proposed LMScore. For each example, the expert was asked to rate a response as "good" if it had few or no errors compared to the ground truth, "okay" if it had many minor errors, and "bad" if there were major errors. We use these human responses to compare and correlate the newly proposed LMScore which is reported in Fig. 10.

Table 5 shows performance of the proposed LMScore (LMS) for all models across all tasks. We observe that LMScore values show trends similar to what is observed in ROUGE-L (R-L). Table 6

# E    LIMITATIONS

This work focused on a select set of domains and a narrow set of tasks with high quality annotations, thus limiting the scale. Increasing the scale of examples across tasks would provide a more robust

| Method | DFT-C | | DFT-S | | | DFT-P | | | DFT-AVG | |
|---|---|---|---|---|---|---|---|---|---|---|
| | R-L | LMS | LLMSim F1 | R-L | LMS | LLMSim F1 | R-L | LMS | R-L | LMS |
| *Zero-shot Open Weight LLMs* | | | | | | | | | | |
| Mixtral | 7.43 | 0.09 | 25.61 | 14.03 | 0.05 | 7.87 | 15.16 | 0.10 | 7.43 | 0.08 |
| Command-R+ | 5.12 | 0.06 | 34.47 | 18.68 | 0.03 | 6.97 | 17.59 | 0.09 | 5.12 | 0.06 |
| LongLLaMa | 4.61 | 0.03 | 1.96 | 8.68 | 0.01 | 5.02 | 11.23 | 0.02 | 4.61 | 0.02 |
| *Zero-shot Closed-Weight LLMs* | | | | | | | | | | |
| Gemini 1.0 Pro | 6.57 | 0.10 | 42.95 | 12.42 | 0.02 | 9.66 | 14.67 | 0.05 | 10.52 | 0.06 |
| GPT-4o | 9.05 | 0.16 | 33.30 | 15.70 | 0.04 | 24.76 | 15.17 | 0.15 | 9.05 | 0.12 |
| Claude 3 (Opus) | 8.38 | 0.16 | 36.28 | 16.87 | 0.05 | 21.81 | 16.10 | 0.13 | 8.38 | 0.11 |
| Gemini 1.5 Flash | 10.07 | 0.22 | 39.34 | 10.96 | 0.05 | 21.36 | 12.90 | 0.13 | 10.62 | 0.13 |
| Gemini 1.5 Pro | 10.83 | 0.21 | 36.99 | 14.55 | 0.03 | 20.89 | 15.59 | 0.12 | 10.83 | 0.12 |

Table 6: Results comparing performance of all models on the DFT tasks based on the ROUGE-L metric (R-L), LLMSim (F1) and the proposed GPT-4o based LMScore (LMS) metric. Blue is highest score and Yellow is second highest.

evaluation benchmark. With the fast pace of language model advancements, evaluating the generated text responses on such complex tasks is challenging even with high quality human annotations. In particular, just based on instructions and output format provided in the prompt, existing automated evaluation metrics Rouge-L and BERTScore can be unforgiving resulting in low scores for responses that look different but might still be reasonable. While we propose model based evaluation metrics, these are still far from perfect and provides room for more creative strategies. Further, we primarily evaluate models in the zero-shot and two-shot settings (discussed in the supplement) and we invite researchers to explore retrieval augmented generation and chained prompting strategies that evaluate the models on planning and task decomposition.

Another limitation is that, it is difficult to obtain human performance on the benchmark. These tasks require extensive expertise and it's unlikely that any one person has sufficient expertise across these domains. On the other hand, hiring an expert for each of the domains and evaluating their performance of the task would double the cost of annotation, so in this work we rely on annotator adjudication to come up with the final answers.

## F  DETAILED DESCRIPTIONS OF TASKS AND DATA

### F.1  DENSITY FUNCTIONAL THEORY (DFT) TASK

Density functional theory (DFT) provides a robust framework for quantum mechanical modeling of materials, enabling first-principles predictions and validation of experimental findings. Despite its widespread use and success in materials science, DFT calculations remain largely inaccessible to most experimental researchers, typically requiring a specialized PhD-level training in computational materials science. This knowledge gap arises from a confluence of factors: the underlying quantum mechanical formalism; intricacies of numerical implementation and software-specific parameters; and the nuanced interpretation of results. While the theoretical foundations are often covered in elective graduate coursework, the practical application of DFT necessitates years of specialized training to develop the intuition required to navigate software complexities, select appropriate parameters and ensure the physical validity of results.

This benchmark represents a critical first step towards harnessing LLMs for automating complex scientific workflows. While considerable interest exists in leveraging AI to accelerate scientific discovery, concerns persist regarding the accuracy and reliability of LLMs in this context. Evaluating the performance of LLMs on intricate scientific tasks, such as DFT calculations, presents a significant challenge due to the domain expertise required for assessment. To address this, we introduce a comprehensive framework for evaluating each stage of the DFT workflow - from initial planning to final execution, ultimately aiming towards the complete automation of DFT calculations. Such automation would empower a broader range of scientists including materials scientists, chemists and physicists to perform complex calculations without the need for extensive specialized training, enriching the theoretical foundations of experimental work and accelerating discovery.

To assess the ability of large language models (LLMs) to generate DFT workflows, we propose a benchmark task requiring the translation of materials science papers into executable Python code, leveraging established domain-specific libraries. This benchmark encapsulates three core subtasks: (1) identifying the computational steps essential for reproducibility; (2) extracting metadata, including input structures and DFT parameters; and (3) translating these steps and metadata into functional code. Subtask (1) evaluates the LLM's comprehension of domain-specific concepts and ability to plan complex procedures. Subtask (2) assesses its capacity to extract pertinent information within a potentially expansive context, where relevant details may be dispersed across different sections of the publication. Finally, subtask (3) gauges the LLM's aptitude for scientific coding which requires not only proficiency in a specific coding language and libraries but also a deep understanding of the underlying scientific principles.

### F.1.1  DFT DATASET COLLECTION DETAILS

We identified a set of $\sim 200$ papers from the S2ORC corpus[1] that mentioned DFT computations in the abstract. We hired two expert annotators, who hold PhD degrees specifically in Materials Science and work directly on DFT computations to annotate the papers. The annotators identified 74 papers which had DFT calculations to validate experimental results and also included a mix of theoretical understanding. Some of the papers that were discarded included papers that were proposing new DFT computation methods, or papers that were large focused on theory with just a passing mention of DFT computations. The annotators were asked to annotate all information necessary to reproduce the DFT computations in the paper.

The annotators started by identifying the structures studied in the paper. They identified the graph of the computational steps carried out along with the inputs and outputs that went into each computation. They used the inputs and outputs to perform a topological sorting of the functions to create a computational graph of the different DFT computations done in the paper. They then implemented the functions of the DFT computation in python code. In cases where the paper did not have sufficient information, they marked such functions with placeholders noting that additional information was necessary and missing from the paper to fully reproduce that specific step. The annotators also identified analysis steps, and include code to perform the analysis functions. The annotators provide final code that includes the structure metadata, DFT params, and general helper functions as well as specific functions for each of the computation steps, and a final block of code that executes the computations and the analysis in the sequence performed in the paper.

### F.1.2  DFT TASK DETAILS

Our dataset is composed of three parts: calculation metadata for input structures, DFT computation functions and parameters, and code.

**Metadata for input structures and DFT functions** We capture the details of the inputs to each DFT calculation run in the workflow in two different types of metadata: structure metadata for the input structure, and DFT parameters for the DFT calculation settings. These subtasks are named DFT-S and DFT-P respectively and prompts are shown in Fig. 11 and Fig. 12.

**Code for reproducing the calculation workflow** We write python code that reproduces the calculation workflow specified by the computation graph and the calculation metadata. The code is written in python, and we primarily use the Atomic Simulation Environment (ASE) library for setting up, manipulating, running and analyzing DFT calculations. We assume that the DFT software used in the original paper are available to run, and call the DFT software from ase. The prompt for this is shown in Fig. 13.

---

[1]S2ORC: https://github.com/allenai/s2orc

**DFT-S Prompt**

```
A materials scientist would like to reproduce the DFT calculations
from a paper.
They want to identify the input structures, and gather as much
information about the structures as possible.

Make sure to identify all input structures, and output a list of
the distinct input structures information with fields
"id", "common_name", "scientific_name", "type", "composition",
"description", "vacuum_x", "vacuum_y", "vacuum_z", "supercell",
"cas_number", "lattice_a", "lattice_b", "lattice_c", "space group",
"orientation", "mp_id", "isomer_name".
The "id" field is just a string of format "structure_metadata_{number}",
where {number} starts from 1 and indicates the order the structure
appears in the excerpt.
The "description" field should capture all relevant information
that is not captured by the other fields.  Make sure to write your
output as a list of dictionaries, NOT A BULLETED LIST.
The "common_name" field should have the common name of the material
and "scientific_name" should have the formal scientific name of the
material.
The "type" field should point out whether the material is a
molecule, a protein, a bulk crystal structure, a thin film or
something else.
If the structure has vacuum around it, please include the thickness
of vacuum layer in three directions with units in these fields:
"vacuum_x", "vacuum_y", "vacuum_z".
The fields "lattice_a", "lattice_b", "lattice_c" correspond to the
lattice parameters of the unit cell.  Include units in these fields
as well.  Leave them blank if lattice parameters are missing.
The "supercell" field should tell how many times the unit cell is
repeated in the [x, y, z] directions, like "2x2x2".
If the text indicates the material structure is from material
project and provides a "mp_id", please include that in the field
"mp_id".
If the structure has multiple isomers and the text mentions which
is used, please indicate that in the field "isomer_name".
If any information is relevant but missing, input "UNKNOWN" in the
field.  If any information is irrelevant and missing, input None in
the field.
The "description" field should capture all relevant information
that is not captured by the other fields.  Make sure to write your
output as a list of dictionaries, NOT A BULLETED LIST.

Example excerpt:
[Excerpt and additional electronic configuration details
abbreviated]

Example output format:
{
"id":  "structure_metadata_1",
"common_name":  "STO",
"scientific_name":  "Strontium titanate",
[Additional fields abbreviated:  type, composition, all parameters]
"description":  "pure SrTiO3" [Additional structures abbreviated]]
}
Here is the paper:
{{text}}

---
Identify the input structures, and gather as much information about
the structures as possible.
Use the format from the example output.
```

Figure 11: Prompt for extracting input structures used in DFT calculations (DFT-S Task).

---

**DFT-P prompt**

```
A materials scientist would like to reproduce the DFT calculations
from a paper.
They want detailed information about each of the DFT calculations
in the paper.
Output a list of the distinct DFT parameters with fields including,
but not limited to, "software", "functional", "k-points-grid",
"pseudopotentials",
"basis_set", "energy_cutoff", "energy_convergence",
"force_convergence", "relaxed_nuclei",
"relaxed_unit_cell", "spin", "hubbard_U".
If a structure is relaxed, please set "relaxed_nuclei" or
"relaxed_unit_cell"
to 1.0 (corresponding to true).
If spin is involved in the calculation, set "spin" to 1.0
Include the units in these fields if applicable.
If any information is relevant but missing, input "NaN" in the
field.
If any information is irrelevant and missing, input "NaN" in the
field.
Make sure to write your output in JSON format.
Include any related information to the that has not been covered
into the "other_information" field.

Use the following format:
{
"function_name":"short_function_description",
"software":  "Specify which software was used, e.g.  vasp,
gaussian, castep, qe, dmol, orca, wein2k",
"functional":  "functional name",
"k-points":  "k-point grid as [x,y,z
",
"energy_cutoff":  "energy cutoff in eV if mentioned, else NaN",
"energy_convergence":  "energy convergence if mentioned, else NaN",
"force_convergence":  "force convergence if mentioned, else NaN",
"relaxed_nuclei":  "1.0 if nuclei is mentioned to be relaxed, 0.0 if
mentioned to be fixed, else NaN",
"relaxed_unit_cell":  "1.0 if it is mentioned to be relaxed, 0.0 if
if mentioned not relaxed or else NaN ",
"spin":  "1.0 if spin considered, 0.0 if not considered, else NaN",
"hubbard_U":  "NaN if not mentioned",
"other_information":  "Any other relevant information for the
calculation."
},
...
]
[Example excerpt abbreviated]

[Example output:  Single DFT calculation with VASP, HSE06, key
parameters shown]

Here is the paper:
{{text}}

Using the specified format, extract and
list out the details of all the DFT calculations in the paper.

Answer:
```

Figure 12: Prompt for extracting DFT calculation parameters (DFT-P Task).

```
DFT-C prompt

I am a computational materials scientist, and I would like to
reproduce the DFT calculations from a paper.
Write python code to reproduce all DFT calculations from a paper.
You have access to the library ase, and any DFT software used in
the paper.
Whenever possible, make the code modular by write functions for
each step of the calculation
(i.e.  set up the unit cell, run DFT, find the total energy from
the DFT calculation, plot the band structure, ...)
and calling the functions, instead of writing one long body of
code.
Make sure all input structures, DFT calculations, and calculation
outputs such as energies, density of states or band gap are
accounted for.
---
This is the paper I'd like to write the python code for:
{{text}}
---
Output:
```

Figure 13: Prompt for generating Python code to reproduce DFT calculations (DFT-C Task).

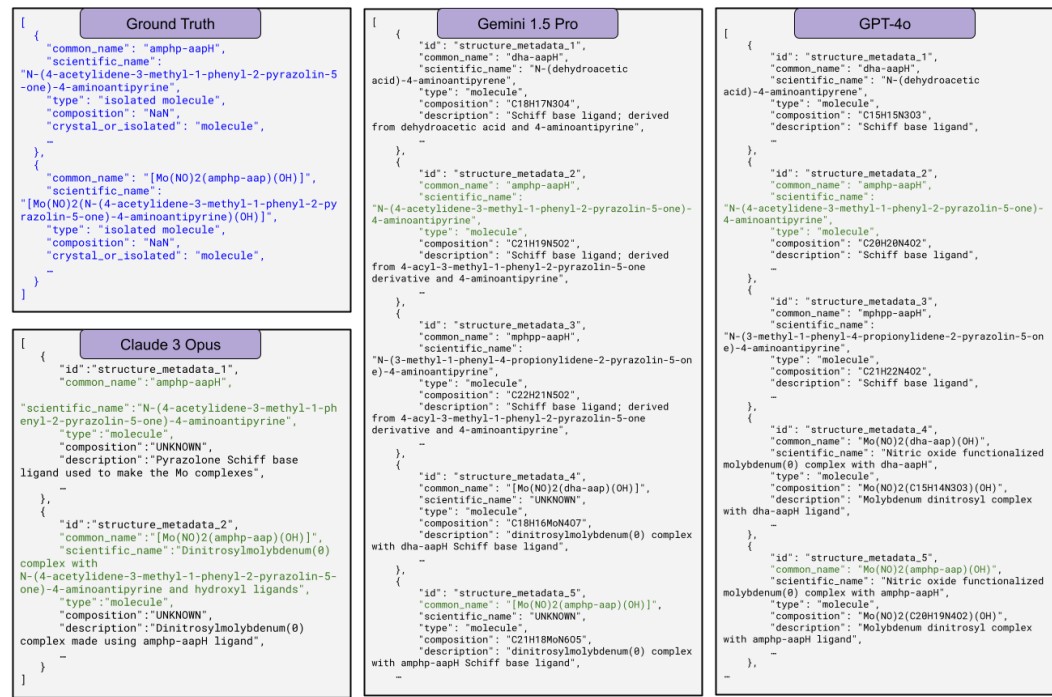

Figure 14: **Model responses on the DFT-S structure metadata extraction task**. Claude-3 Opus extracts accurate structures (green) relevant to the DFT computation whereas the other models do not precisely identify the exact structures that go into the DFT computation and tend to repeat entries.

### F.1.3 DFT TASKS: RESULTS AND ANALYSIS

**Structure Metadata. (DFT-S)** Many of the models did a good job of picking out relevant information about the chemical structures from the text. Qualitatively, the claude-3-opus-20240229 model outperforms the others in its ability to understand how many structures are described in the paper. For example, on record 2023-09-22-13bcf90c3ef43f1413deg, claude-3-opus-20240229 correctly identifies two structures that are present in the text. The other models output many more struc-

tures with repeated information. The same is true for records 2023-09-22-01b9cdba467fd7882e42g and 2023-09-22-07b4d66e23971ccb85c0g

**DFT Paramaters Metadata. (DFT-P)** Again, all models do a reasonable job of identifying relevant information from the text and structuring it properly as prompted. However, claude-3-opus-20240229 appears to understand the purpose of the calculations better than the other models, and it avoids unnecessary repetition For example, in record 2023-09-22-07b4d66e23971ccb85c0g, claude-3-opus-20240229 correctly identifies that there is one set of DFT parameters used in the actual study as well as two more sets of parameters used for convergence testing. Based on how it names each parameter set, it is clear that it understands the purpose of each parameter set. For example, it named a parameter set "DFT-parameters-convergence-tests" as opposed to the other models' more generic names, like GPT-4o's "create-dft-parameters-rpbe" However, its predisposition to brevity has its drawbacks as well. For example in record 2023-09-22-0ce1b5ea9a8637db5435g, claude-3-opus-20240229 incorrectly finds only one set of DFT parameters. Meanwhile, gemini-1.5-flash-latest correctly identifies the two sets of parameters actually present in the paper. All other models reproduce the same sets of parameters multiple times with different names.

**Code (DFT-C)**. Gemini-1.5-flash-latest and mixtral-8x7b-32768 do much worse than the others. E.g. record 2023-09-22-0ce1b5ea9a8637db5435g. Gemini-1.5-flash-latest returns simple constant values like "return 0" and "return [0]" where other models implement more useful functions. Mixtral appeared not to finish writing a script at all. The others arguably performed similarly. Claude3 was the best overall.

## F.2 MATERIAL, PROPERTY, VALUE EXTRACTION TASK (MPV)

Modern scientific research articles contain unstructured natural language descriptions of materials, their structure, processing, and properties. This wealth of information is often difficult to access due to the unstructured nature of the text. Collection and standardization of such unstructured information at scale would benefit materials researchers and accelerate new materials discovery. Existing resources, such as human-curated tables or handbooks, are typically limited in scope and often lack crucial details regarding material structure and processing. This lack of context can lead to poor predictability of material properties. While rule-based approaches have been widely employed Swain & Cole (2016); Mavracic et al. (2021b); Dong & Cole (2022b), they often fail to capture critical information hidden within the broader context of the text. Dataset curated by computational methods is a good alternative with structured information, however computational results could not be generalized to the prediction of experimental measurements mp (2017) due to the inherent limitation in the theory itself (e.g. density functional theory (DFT)).

With LLM, prompt-based extraction from the scientific literature is emerging as a new extraction method that could result in better accuracy at a lower amount of human effort. Early LLM-based property extraction studies utilized fine-tuning Dunn et al. (2022), but there have been a series of papers using prompt engineering with success, such as extraction of critical cooling rates of metallic glasses, yield strengths of high-entropy alloys, emission wavelengths of phosphors or synthesis parameters metal organic frameworks Zhang et al. (2023); Zheng et al. (2023); Polak & Morgan.

### F.2.1 MPV: DATASET COLLECTION DETAILS

We identified a set of 17 papers from the S2ORC corpus[2]. We hired expert annotators to annotate the papers and extract all material names, material descriptors, property studied along with property descriptors, and values corresponding to the properties as mentioned in the paper. The annotators were also asked to provide the source sentence to help validate the extractions.

### F.2.2 MPV: TASK DETAILS

We perform 3 sets of tasks.

**Extraction of all properties**. We ask the model to extract all materials, properties and values mentioned in the paper. Prompt for this is shown in Fig. 15. This is identical to the task description given to the annotators. This task also requires that the final output includes the material descriptor, property descriptor and source sentence corresponding to the descriptors and values being extracted. This is the main MPV task reported in the results in Table 1 and Figure 5.

---

[2]S2ORC: https://github.com/allenai/s2orc

---

**MPV prompt**

```
You are a materials scientist.  Your goal is to find and extract
all numeric values or numeric value ranges of material properties
mentioned or tabulated in a given paper text.

The output should be in JSON format:
[
{
"material":  "<material name>",
"material_descriptor":  "<material descriptor>",
"material_source_passage":  "<source passage from which the material
name and descriptor were identified>",
"material_source_table":  "<source table from which the material
name and descriptor were identified>",
"property_name":  "<property name>",
"property_descriptor":  "<property descriptor>",
"property_source_passage":  "<source passage from which material
name, property and property descriptor were identified>",
"property_source_table":  "<source table from which material name,
property and property descriptor were identified>",
"low_value" :  "<low value>",
"high_value":  "<high value>",
"value_units":  "<value units>",
"value_source_passage":  "<source passage from which material
property values were identified>",
"value_source_table":  "<source table from which material property
values were identified>",
}, ...
]
An example of the output is as follows:
[Example:  Single entry for HfO₂ with refractive index 1.84]

There are some certain rules you need to follow:
1.  Be thorough!  Don't miss even a single numeric value.
2.  Avoid null low_value and high_value.  Simply skip the entry if
you cannot extract the numeric value.
3.  If a value is only extractable from a figure, not text or
table, simply skip that entry.
4.  Make sure whatever low_value and high_value actually exists in
the source passage or referred table.
5.  If one passage contains multiple materials or properties,
record all of them as separate entries.
6.  If different values are mentioned for the same material
property, record all of them as separate entries.
7.  Material, property, and value may or may not be mentioned in
the same passage.

This below is the excerpt from a paper in LaTeX format published
in a materials science journal.
Excerpt:
text:
{{text}}
------------------------------
With the above excerpt, the goal is to extract numeric values or
numeric value ranges of material properties from the above paper.

Please only output your answer in the exact format as shown above
without any prefix.
Output:
```

Figure 15: Prompt for extracting material property values from text (MPV Task).

**Extraction of specific properties.** The second task we do is to test performance when we narrow down specific properties of interest. We specifically ask the model to extract materials, property values where refractive index and bandgap properties are being studied. This is a more tightly scoped task, and we do this specifically to evaluate the models on a narrower well defined task. As with the task of extracting all material property values, we ask the model to also include the material descriptor, property descriptor and source sentence corresponding to the descriptors and values being extracted.

**Extraction of non-trivial properties.** We also evaluate a minor variation of the first task where we ask the model to extract all non-trivial properties of materials and their values. These properties in some sense would be considered to be important contributions of the experimental work reported in the particular research paper. This is to avoid any confusion between generic properties of materials that are known prior to a given research paper vs. those specifically studied or identified newly in a research paper. This task is useful in extracting just the additional information necessary to populate a database of materials, properties and values.

### F.2.3 MPV: DIFFICULTY RATING LOGIC

When rating the difficulty of each example for the MPV task, the experts marked an example as "Easy" if the answer could be found in the original sentence without modification. Examples were marked "Medium" if the answer was not very obvious, and the descriptors are very important and cannot be missed. Examples were marked "Hard" if the answer was derived from multiple sentences, and sometimes from tables and figures and reasoning was necessary to determine the right descriptors or values.

Figure 16: Example from the MPV task where Claude-3 Opus, Gemini 1.5 pro, and GPT-4o recalled some important material properties and values but failed to capture all relevant properties and focused instead on some trivial properties. (paper id: 53519111)

### F.2.4 MPV: RESULTS AND ANALYSIS

Both precision and recall rate were used in the evaluation process. A tuple of material, material descriptor, property, value were used for matching and then measuring if property value is accu-

Figure 17: Example from the MPV task where all models correctly identify and extract a material, property and other information but have slight differences. (paper id: 15804005)

rate. The model output and ground truth are only considered as a match if these four fields are semantically equivalent. Exact match is often impossible because of the existence of alternative names for material and properties. The material descriptors could also be part of the material name, which makes the exact match very difficult. For example, "$TiO_2$ nanocrystal" as a material name is equivalent to "$TiO_2$" as material name and "nanocrystal" as material descriptor.

In the context of model evaluation, we once again incorporate LLM models into the workflow in order to analyze the level of correspondence between the material properties extracted by the previous LLM and the material properties annotated by humans. This process is referred to as the model evaluation. During the model evaluation, both the material properties annotated by humans and those extracted by the LLM are fed into the LLM. Prompt engineering is utilized to enable the LLM models to identify matches based on specific attributes: material names, property names, low and high values, and unit of measurement. To ensure comprehensive matching, we specifically instruct the model to exhibit leniency; this allows it to encompass properties that, despite lacking identical names, convey the same meaning (e.g., Indium Nitride vs. InN) or differ in the order of units (e.g., $100cm^2$ vs. $1 \times 10^2 cm^2$). Furthermore, we emphasize the importance for the LLM to pay attention to descriptors of materials and properties, as these may indicate a significant difference in the form or state of the material. During the model evaluation process, the source passages are not taken into consideration while instructing the LLM. This decision is made because the same properties can be extracted from different passages. Additionally, it is not uncommon for different annotators to select different sections of the same passages or to use different lengths as their source passages. Consequently, the source passages are only utilized as a reference for human attention when identifying and analyzing issues of evaluations.

By correlating each human-annotated material property with a matching or empty LLM-output property, we can compute the precision and recall of the model evaluation, as illustrated in Table 2.

The results of the model evaluation provide valuable insights into the potential transformation of the entire task into fully automated pipelines all by LLM, with the LLM doing both the material property extraction and extraction result evaluation tasks.

It is also worth mentioning that the LLM outputs also include trivial material-property entities such as sample thickness. Human annotators omit this information due to the lack of general interest to materials science community. Thus we explicitly asked LLM to only include non-trivial material property entities in the outputs and the precision is significantly improved while the recall rate is similar, as shown in the "mpv-non-trivial" column of Table 3. We further limit the properties to "bandgap" and "refractive index" as these two properties are the most common material properties reported in scientific literature. The results can be seen in the "mpv-specific" column of Table 3: the precision is significantly higher than the regular mpv task due to less ambiguity on whether a property should be included. But the recall rate is reduced, which is also related to the filtering we applied in the prompt.

### F.3 HARTREE-FOCK MEAN-FIELD THEORY TASKS (HFD, HFE)

In this section we provide additional details about the datasets, detailed task description and analysis of the results for the condensed matter physics tasks. Condensed matter physics often utilizes the Hartree-Fock method, which involves deriving a Hartree-Fock (mean-field) Hamiltonian.

#### F.3.1 HFD AND HFE TASK DETAILS

We select two tasks pertaining to Hartree-Fock mean-field theory.

**Hartree-Fock Derivation (HFD).** The first task, HFD, involves analytically deriving the Hartree-Fock Hamiltonian through a series of 13 to 19 intricate steps. Each step demands a deep understanding of the physical system and requires specific mathematical operations. Due to the complexity of this task, we employ a template approach. The template provides a general structure for each step and includes placeholders for key components. The specific task for the Large Language Model (LLM) is towards filling these placeholders which would allow for analytic derivation of a Hamiltonian at each step. We evaluate the LLM's performance by comparing the responses for the placeholders with the ground truth written by an expert. Aside from filling in the placeholders at each step, we also ask the model to derive a Hamiltonian at each step and compare it to the ground truth Hamiltonian. The entire prompt used for this task contains about 6k words.

**Hartree-Fock Extraction (HFE).** Recognizing the challenges inherent in HFD, we introduce the Hartree-Fock extraction task (HFE) as a simpler comprehension task as opposed to the deep reasoning required in HFD. This task involves extracting a Hartree-Fock Hamiltonian from scientific papers that contain this Hamiltonian. If the Hamiltonian contains some terms that are defined in the paper, we ask the model to extract those terms as well. While HFE is less demanding than HFD, it presents its own challenges, as a single paper may contain multiple Hartree-Fock Hamiltonians specific to different cases being studied. So we ask the LLM to fill in a template with an exact Hamiltonian expression and all the terms that are necessary to describe it. We note that a Hamiltonian, in most cases, contains several terms that are written in different parts of the paper. For example, it usually contains non-interacting term ($H_0$) as well as interacting $H_{int}^{HF}$ terms. While the non-interacting term is usually unique, the interacting term can take on various forms in a given paper and, therefore, the LLM must choose which interacting Hamiltonian is the most general.

By dividing the process into these two tasks, we can systematically evaluate the capabilities of LLMs in tackling complex physics problems. HFD allows us to assess the model's ability to perform step-by-step complex, scientific analytical derivation, while HFE focuses on reasoning and information extraction. The prompt for HFE task is shown in Fig. 18 and a condensed version for HFD in Fig. 19.

#### F.3.2 HARTREE-FOCK DATASET COLLECTION

We have two datasets for the two tasks (Hartree-Fock derivation and Hartree-Fock extraction).

```
HFE prompt

You are a physicist.
You are reading a paper that has explicit equation (or equations)
for the general Hartree-Fock or mean-field Hamiltonian.
There are might be several Hartree-Fock Hamiltonians in the paper.
You should return the one that is the most general.
Return this Hamiltonian.
Print out each equation explicitly instead of citing it.
Print out terms in the Hamiltonian if they are present in the
paper,
including intergrals.
Do not explain the Hamiltonian or the terms
Return the Hamiltonian in the following format:
'The general Hartree-Fock Hamiltonian is
{{The Hartree-Fock or mean-field Hamiltonian}}
where {{include all terms in the Hamiltonian}}'
Be concise.  Do not explain constants.

PAPER:
{{text}}

YOUR RESPONSE:
```

Figure 18: Prompt for extracting the general Hartree-Fock Hamiltonian from a paper (HFE Task)

**HFD.** The dataset for the HFD task consisted of the papers used in Pan et al. (2024). These were hand selected by expert post-doctoral scientists intimately familiar with the derivations in the work. 15 papers were selected and reasoning steps from solving the quantum many body system was identified for each paper. Pan et al. (2024) also provides prompts and includes derivations to compare responses from a language model. The contribution of our work is a generalized version of the template created by Pan et al. (2024). We create a format that is consistent across all papers so that evaluation can be automated. The ground truth for task includes all of the derivations associated with each of the reasoning steps annotated in Pan et al. (2024).

**HFE.** The papers for the HFE dataset were selected from those that contained Hartree-Fock Hamiltonian, discovered using the arXiv advanced search. Specifically, we performed advanced search on arXiv with the following parameters: $(i)$ We set Classification: Physics: Condensed Matter; $(ii)$ we set include cross list as 'True'; $(iii)$ we look for 'Hartree-Fock' as key words in abstract. We then selected papers in reverse chronological order (from most recent onwards) and filtered to get a total of about 50-80 papers. An expert doctoral candidate in the field then filtered papers where the Hartree-Fock Hamiltonians were present in the paper directly and also extracted and aggregated the equations form across the papers to create a ground truth evaluation set. The final dataset for the HFE task consisted of 38 papers.

### F.3.3 DIFFICULTY RATING LOGIC

Two experts marked each example with the difficulty ratings using the following rubric.

**HartreeFock Extraction task.** Easy: If the non-interacting term $H_0$ and interacting terms $H_{HF}$ are on the same page or in the same section and the paper does not have many formulas, so the Hamiltonian is easy to find then it was marked "easy". Medium: If $H_0$ and $H_{HF}$ are in different parts of the paper the example was marked "medium". Hard: If the paper contains many formulas, or several HF Hamiltonians and it requires time or involves logic to find the right Hamiltonian in the paper then it was marked "hard".

**HartreeFock Derivation task.** For the derivation task, the experts evaluated the non-interacting term and the interaction term separately as either "easy" or "hard". If both are "hard" then the example was marked as "hard". If both are "easy", then the example was marked as "easy". Otherwise, the example was marked as "medium".

---

**HFD prompt**

You are a materials scientist. You are provided with a paper and the STEPS to derive a Hartree-Fock Hamiltonian.
**Be concise. Some STEPS are optional. You should decide whether or not to perform the step based on your understanding of the paper.**

In the STEPS below, you will need to answer the questions or deduct necessary information from the paper in order to fill in the placeholders {}.
You should print a Hamiltonian at each step whenever it is requested to derive.

**Placeholders Explanation:**
1. The {} placeholders are intended for string substitutions.
2. The {A|B} placeholders are intended for string substitutions with either A or B.
3. The {text|None} brackets denote optional strings (text).

**System Choice Before Derivation:**
Before starting the derivation, you should make a choice among these three possibilities for the system based on the paper:
This is very important. The choice is strongly correlated with the description of the problem in the paper.

Which Hamiltonian does the paper study? Choose one:
1. Continuum version, single-particle
2. Continuum version, second-quantized
3. Lattice version

*Note: Do not print out the choice. Rather, follow the steps based on the choice.*
**Output Format:**
Respond in a valid JSON-formatted string. Ensure your response follows this format:

```
[
  {
    "Step": "1",
    "task": "Construct Kinetic Hamiltonian",
    "answer": "Hamiltonian ..."
  },
  [Additional steps follow the same format]
]
```
**STEPS:**
**Step 1: Construct Kinetic Hamiltonian**
Express Hamiltonian using proper operators and summations.

**Step 2: Define Terms in Kinetic Hamiltonian**
Choose dispersion: parabolic, Dirac, or cos-like.
*[Example dispersions abbreviated]*

**Step 3: Construct Potential/Interaction Hamiltonian**
Add diagonal and off-diagonal terms.
*[Detailed potential terms abbreviated]*

**Step 4: Convert to Second Quantized Form**

**Step 5: Apply Wick's Theorem**
*[Additional steps and examples abbreviated]*

**Here is the paper:**
{{text}}

Figure 19: Prompt for deriving the Hartree-Fock Hamiltonian (HFD Task).

Figure 20: Model responses for an example in the HFD task.

### F.3.4 HF TASKS: RESULTS AND ANALYSIS

**HFD.** As described in the main text, the derivation task is challenging, and we expect poor LLM performance. Fig. 5 shows that Gemini 1.5 Pro performs better than other models. Gemini 1.5 Flash, Gemini 1.0 Pro, Claude 3, and Command R Plus also achieved good scores. GPT-4o and LongLLaMa performed the worst. We note that GPT-4o's score was affected by a formatting issue. Although we instructed the model to provide the output in JSON format, GPT-4o's output included the words "key" and "value," which were not present in the ground truth. Despite this, the average score for the HFD task across all models is much lower than for the HFE task, which a human expert considers simpler.

**HFE.** The extraction task does not involve extensive instructions and is considered simpler than HFD. As shown in Fig. 5, all models (GPT-4o, Claude 3, Gemini 1.5 Flash, Gemini 1.5 Pro, Gemini 1.0 Pro, and Command R Plus, listed in descending order of performance) performed consistently well on this task. However, LongLLaMa performed significantly worse than any other model. Overall, the average score for this task is much better than the score for the HFD task, as expected. This difference in performance is even more evident when using the proposed LMScore as seen in

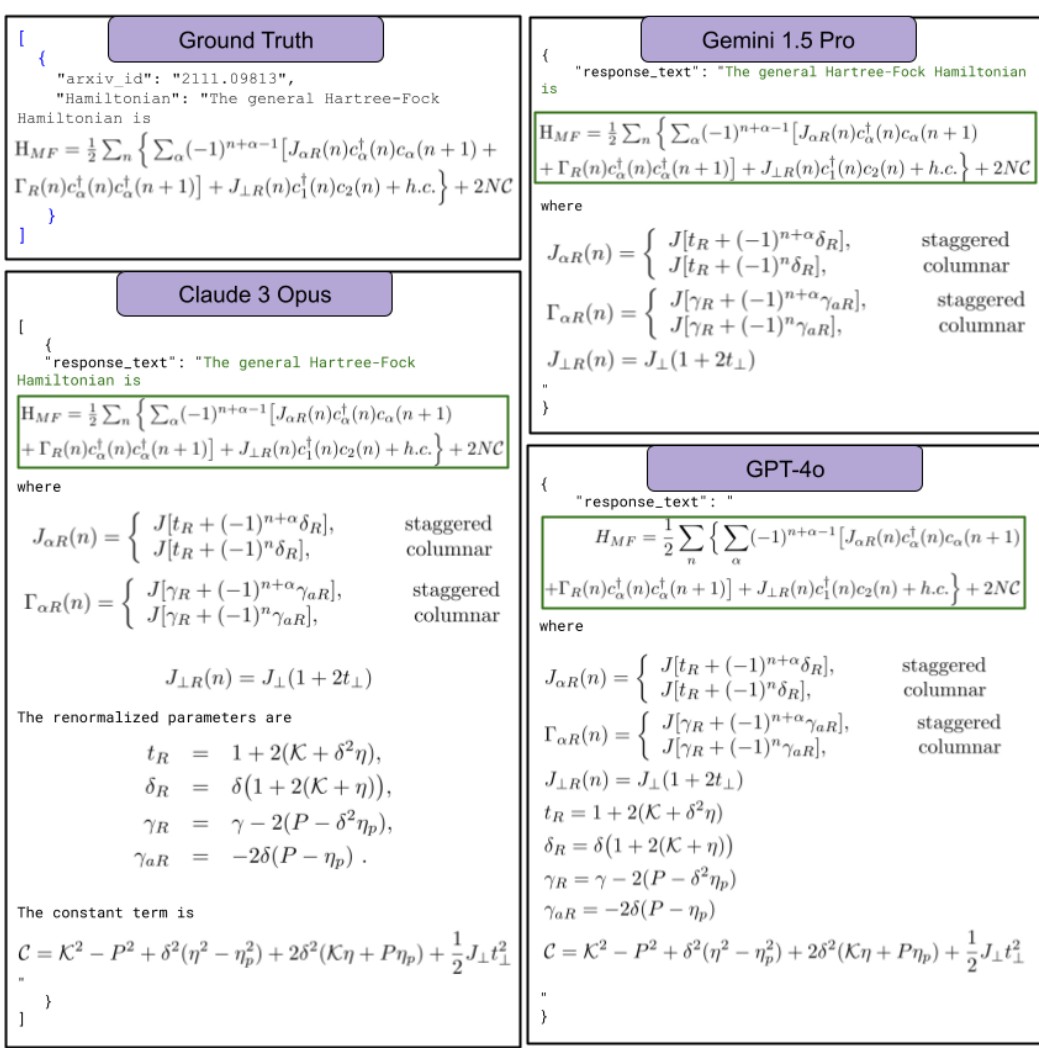

Figure 21: Model responses for an example in the HFE task.

Fig. 10, which presents the human evaluation of LLM performance, showing better results for the HFE task than the HFD task.

## F.4 (QUANTUM) ERROR CORRECTION ZOO TASK (QECC)

The Error Correction Zoo (EC Zoo)(Albert & Faist, 2024) is an open source effort to build a Wikipedia-like repository collecting and categorizing error correcting codes from the literature. Such codes are used to redundantly encode and store classical or quantum information so as to protect it from noise, with wide applications (e.g., 5G communication and broadcast protocols, quantum communication and computing). Currently, the construction and maintenance of the zoo is carried out manually by a "Zookeeper", who updates the EC Zoo's contents whenever a paper introducing a new code appears on the arXiv repository.

An entry in the EC Zoo consists of a YAML file, listing the properties of a given EC code along with any relations to other codes. Each entry contains many different types of information: a succinct non-technical summary, bespoke technical details, numerical parameters quantifying code performance, and non-trivial connections to other literature. Due to the diversity of encoding schemes, code entries can be very different, and there is no one-size-fits-all template. We construct a benchmark that tests the ability of LLMs to curate the EC Zoo.

### F.4.1    QECC: Task Details

The goal of the QECC task is to produce the YAML file similar to the EC Zoo code entry given a paper and an appropriately designed prompt description of the task. The models output is then compared against the human-curated ground truth. Prompting an LLM to provide such an entry tests its ability to correctly extract both the relevant information Mayfield et al. (2024) and the context.

Figure 22 shows the prompt for this task which lists various available YAML key-value pairs taken from a template code YAML file[3].

### F.4.2    QECC: Dataset Collection

We curated the benchmark from two experts in the domain who have prior experience curating entries for the ECZoo. Our benchmark consists codes from 65 arXiv papers ranging from 1997 to 2024. Each paper introduces a particular code or code family, running the gamut of classical and quantum coding theory and including codes relevant to high-energy/particle physics, phases of quantum matter, theoretical computer science, electrical engineering, and quantum engineering. More specifically, the 65 codes include 15 codes realizing phases of quantum matter, 14 codes designed for continuous-variable quantum systems, 12 high-performing codes designed for qubit-based quantum devices, 7 general qubit and 4 general qudit code families, 4 classical codes, 2 fermionic codes, 2 randomized code constructions, 2 codes for transmission of classical information over quantum channels, 2 codes designed for magic-state distillation, and 1 code describing holography.

### F.4.3    QECC Results Analysis

All LLM-generated error-correcting code YAML files were evaluated by the Rouge metric, and a sample of about 20 was carefully read by a human expert. The Rouge metric indicates that LLMs performed relatively well overall.

The human-expert reading of the LLM-generated files affirms that LLMs can effectively summarize technical scientific papers. The summaries tend to be succinct, while at the same time flagging a multitude of key informational "nuggets" and quantitative metrics relevant to a code. The required YAML formatting rules tended to be respected. The performance was consistent across the sample and independent of the underlying discipline of the paper.

In some cases, an LLM flagged informational "nuggets" that were not (but should have been) in a given ground-truth EC Zoo entry. For example, an LLM mentioned a numerical parameter relevant to the performance of a code that was buried in the paper and that was missing from the ground truth. As another example, an LLM pointed out that a particular code was independently discovered by another cited work. Such statements could only be obtained upon a very careful human reading, which takes much longer than running the text through an LLM.

On the other hand, since useful nuggets are not always (and need not be) technically correct, correction from an expert is necessary at this stage. The observed advantage of LLMs is that manual correction is faster than a manual identification of all relevant nuggets.

The level of technical detail in the output files tended to be low, with many LLMs tending to avoid recasting technical details of the paper in pedagogical fashion (see Sec. F.4.4 for an exception). It remains unclear whether this can be mitigated by prompt engineering or if further training is required.

Much of the text in a typical LLM-generated entry consisted of extraneous information that ranged from superficial "fluff" to complete hallucination. In this case, this information was relatively easy to spot by the expert and so did not impede evaluation efforts. It remains to be seen whether this ability to fish out relevant nuggets from the sea of boilerplate holds true for other scenarios.

In summary, current LLMs are beneficial for synthesizing literature for the EC Zoo. More often than not, the LLM thoroughly flagged key aspects of a paper and left only the (faster) process of verification to the human. Running a paper through an LLM produces a draft EC Zoo entry whose

---

[3]https://github.com/errorcorrectionzoo/eczoo_data/blob/main/template.yml

---

**QECC prompt**

```
Fill in a YAML file for the code described in the attached paper
according to the prescription defined in the YAML template below.
Fields are to be filled only if they are directly relevant to the
code introduced in the paper.  Be sure to extract any quantitative
data like thresholds and code rates.  Above all, be concise!  If
you cannot explain something technical in detail, do not try to
explain it.  If something is not detailed in the paper, do not
mention it.

#####################################################
This is a code entry in the error correction zoo.
https://github.com/errorcorrectionzoo
#####################################################

# UTF-8 encoding, AMS-TeX in
( ...
), cite with
cite{arXiv:#.#}
# [Additional documentation comments abbreviated]

code_id:  no_spaces_lower_case # lowercase only
physical:  qubits # one of:  bits, qubits, qudits, etc.
logical:  qubits

Code_parameter:  '((2^r-1,r,d))_{6}' # e.g., ((n,K,d))

name:  'Code-name'
introduced:  '
cite{doi:...}'

description:  |
Brief description, no references.
[Additional description paragraphs abbreviated]

protection:  'Protects against ...'

features:  # Include only if specifically mentioned
[Multiple encoder examples abbreviated to:]
encoders:  ['Process description']
[Multiple decoder examples abbreviated to:]
decoders:  ['Syndrome measurements']
threshold:  ['Error rates']

relations:  # Optional
parents:  [code_id:  parent_code]

_meta:
changelog:
- date:  'YYYY-MM-DD'

Here is the paper
{{text}}
```

Figure 22: Prompt for generating YAML code entries in the Error Correction Zoo Task (QECC).

comprehensive part can be verified and whose superficial part can be deleted, with both jobs done relatively quickly for this task scenario.

### F.4.4 QECC TYPICAL MODEL RESPONSE EXAMPLES

As we analyze the typical model responses, we also present components of the code entry to give a sense for the aspects considered by the human expert when evaluating the model responses. We

go through the output for the code described in Ref. Menicucci (2014). The output below is Gemini 1.5 Pro, with GPT-4o, Claude 3, and Gemini 1.5 Pro producing similar outputs for the same input. Inputting other papers yields outputs with similar features as those covered below.

The header of the output file is generally formatted in the correct YAML format, containing a guess of the code_id key, the underlying physical alphabet used to construct the code, the alphabet used for the logical information, and the code name (with some superfluous information removed). Almost all LLMs respect the desired key-value YAML format throughout the file, and we will not mention this from now on.

```
...
code_id: cv_cluster_gkp
physical: qubits (encoded in continuous-variable oscillators)
logical: qubits
...
name: 'GKP-encoded qubits on CV cluster states'
introduced: '\cite{arXiv:1311.3538}'
```

Next comes a concise description of the code, a typical strength of LLMs. The summary is succinct and well done, while being sufficiently vague so as to not be incorrect. Two quantitative informational nuggets are flagged: the code's 20.5dB threshold and 12.6% error probability.

```
description: |
This code describes a fault-tolerant measurement-based quantum
computation scheme using continuous-variable (CV) cluster states
and GKP-encoded qubits.

The scheme relies on concatenating GKP error correction with a
qubit-level error-correcting code. If the initial squeezing in the
cluster state and the GKP states is above a threshold (20.5 dB for
a typical threshold of 10^-6), the error rate on the encoded qubits
falls below the fault-tolerance threshold of the concatenated
code, enabling fault-tolerant computation.

Magic state distillation of encoded Hadamard eigenstates with an error
probability of 12.6% and success probability of 2/3 allows for
universal computation.

protection: 'Protects against Pauli noise introduced by finite squeezing
in the CV cluster state and GKP encoding.'
```

Next, the LLM produces a list of various features of the code, as instructed. The features to be listed depend on what the paper covers, but LLMs tend to provide text for all available features, irrespective of whether they are discussed in the paper or not. If the paper discusses a particular feature, the LLM flags the relevant information, e.g., the code's 20.5 dB threshold. If the paper does not mention a feature, LLMs tend to put boilerplate for the value in the corresponding key, which ranges from a summary similar to those above to a hallucination. This effect is likely to be mitigated by prompt adjustments.

```
features:
encoders:
- 'GKP encoding using sharp Gaussians.'
transversal_gates: 'All Clifford gates are transversal.'
general_gates:
- 'Universal gate set achieved through magic state distillation of
encoded Hadamard eigenstates.'
decoders:
- 'GKP error correction using ancilla-based measurements.'

fault_tolerance:
- 'Transversal Clifford gates are fault-tolerant when the initial
```

```
squeezing is above the threshold.'
- 'Concatenation with a qubit-level error-correcting code
provides fault tolerance for the entire computation.'

threshold:
- '20.5 dB squeezing threshold for fault-tolerant computation with
a typical concatenated code threshold of 10^-6.'

notes:
- 'The squeezing threshold can be generalized to include
additional noise sources like photon loss and detector
inefficiency.'
- 'The scheme presented is not necessarily optimal, and further
research may lead to better encoding methods and improved
thresholds.'
```

Finally, the LLM presents relations between the primary code extracted from the paper and other codes. Relations form a large portion of the EC Zoo and are difficult to spot because they may be mentioned only briefly somewhere deep in the body of the paper. Using LLMs to flag them saves reading time. In this case, both notable relations are extracted.

```
relations:
parents:
- code_id: cv_cluster_state
detail: 'CV cluster state is the underlying physical resource.'
cousins:
- code_id: gkp_code
detail: 'GKP code is used for encoding qubits in CV oscillators.'
```

**Attempt at technical details**

All LLMs tend to not delve into technical detail, so pedagogy was rare. A notable exception is this attempt by Claude 3 to technically explain codes from Ref. Camara et al. (2007). We present the value of the description key, which includes subsections and LaTeX-formatted equations.

```
description: |
The authors present two methods for constructing quantum LDPC codes.
Both approaches are based on representing the generators of the
stabilizer group as a graph with simple local rules that ensure
commutativity.

\subsection{Generic $(a,b)$-regular construction}
The generic construction produces $(a,b)$-regular quantum LDPC codes,
meaning each qubit is involved in $a$ parity checks and each
parity check involves $b$ qubits. It starts by choosing a group $\mathsf{G}$
and two subgroups $\mathsf{H}$ and $\mathsf{K}$. The cosets of $\mathsf{H}$
correspond to qubits and the cosets of $\mathsf{K}$ to parity checks.
Generators of $\mathsf{G}$ are divided into sets $G_\omega$
and $G_{\bar{\omega}}$, with certain commutation and overlap
properties, to define the Tanner graph.

\subsection{Construction of a $(4,8)$-regular code}
The $(4,8)$-regular construction uses a 3-step procedure. First it
constructs a 4-cycle graph as a Cayley graph. Then a $(4,8)$-regular
Tanner graph is obtained based on the 4-cycle graph. Finally, the
edges are labeled to satisfy the commutation relations. Each qubit
is involved in two $\omega$ and two $\bar{\omega}$ parity checks.
```

F.5    GEOSPATIAL DATASET EXTRACTION TASK (GEO)

A key aspect of geospatial analysis is integrating various diverse datasets together to answer complex questions. For example, a study of time-series snowmelt detection over Antarctica may combine

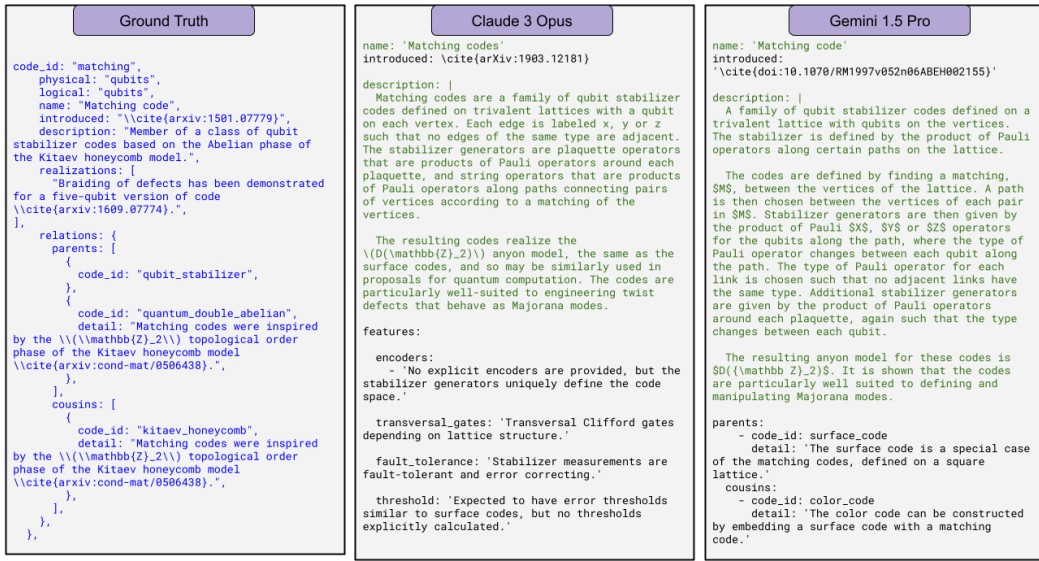

Figure 23: Examples of outputs on the QECC task. All models do a reasonable job in extracting and describing the code. Claude-3 does a good attempt to recover the technical details. (paper id: 1501.07779)

satellite imagery, radar data, weather station temperature data, elevation/topography information, etc Liang et al. (2021). An epidemiological study on infectious disease spread may combine data from the CDC with people dynamics and behaviours (e.g., census, zoning, mobility patterns, Google Trends searches, etc.) Seifter et al. (2010); Zhang et al. (2017). Geospatial analysts leverage their extensive domain expertise to choose representative datasets and then build models to achieve new insights or conclusions that can be used to inform policy, analyze economic or environmental impacts, etc. The goal of our GEO task is to help evaluate how well LLMs can assist with Geo-Analyst workflows.

### F.5.1 GEO TASK DETAILS

In this task we study the ability of LLMs to extract all the relevant information regarding datasets used in a given research paper, including source websites, variable names, descriptions, time ranges and spatial ranges. This task presents a significant challenge to LLMs because it requires not only recognizing direct dataset references within the text, but also contextualization, comprehension and aggregation of information scattered throughout the paper, pushing the boundaries of long-context understanding. Prompt for this is shown in Fig. 24.

### F.5.2 GEO DATASET COLLECTION

Our benchmark consists of 19 papersranging across earth observation, economics, epidemiology and public health, along with ground truth annotations describing the datasets used in each study. To source this dataset we pulled from various sources and repositories (e.g., DigitalCommons, ScienceDirect, MDPI, NIH, Arxiv, etc.), and included only articles with open licenses (e.g., CC BY 4.0, CC BY-NC 4.0, MDPI OpenAccess, etc.) While we started out with more papers, not all of them had permissive licenses thus resulting in a smaller set of 19 final papers. We wanted to represent a variety of domains, as well as varying degrees of dataset extraction difficulty and literature quality. For example, some studies had only a few datasets or variables and others many, some had complex time ranges like "Jan-Apr for every year from 2012-2015", some had a mix of spatial ranges like "every state in the United States, and also 9 countries in Western Europe", some articles were well written and others quite poor, some had information about the datasets cleanly written in one section and others had the information scattered throughout the introduction, problem statement, analyses, and even figure captions, some articles were missing key pieces of information altogether (e.g., regarding time range, spatial range, or source), but a knowledgeable domain expert can correctly guess what these should be if they were familiar with the datasets and study topics.

```
GEO prompt

Given the paper, please gather the following information and put it
in a JSON format.
Here is the JSON format:

{
"paper_title":  <paper_title>,
"paper_link":  <paper_link>,
"datasets":  [{
"dataset_name":  <dataset_name>,
"dataset_website_or_source":  [<dataset_website_or_source>],
"variables":  [{
"variable_name":  <variable_name>,
"description":  <description>,
"time_range":  {
"start_date":  <start_date>,
"end_date":  <end_date>
},
"spatial_range":  <spatial_range>
},],
}],
"notes":  <notes>
}

<paper_title> is the paper title.
<paper_link> is the paper link.
For ALL datasets used in the paper:
<dataset_name> is the name of dataset.
<dataset_website_or_source> is a link to the dataset.
<variables> is the list of variables used in dataset.  Be thorough
and descriptive.
<variable_name> is a list of all the names of variables.
<description> is the description of variable.
* Example:  if data includes all tweets that include a set of
keywords between two dates, explain this with enough detail that
the dataset could be reproduced if we had access to the raw tweets.
For example if the dataset is based on google trends, you need to
include search terms or categories used
* FORMAT: VARIABLE_1, VARIABLE_2, VARIABLE_3,
<time_range> is a list of all time ranges of the variables.
<start_date> is the start date of the time range (format:
yy-mm-dd).
<end_date> is the end date of the time range (format:  yy-mmd-dd).
* Note:  Make sure to give all the time ranges in a list format.

<spatial_range> is a list of all spatial ranges of variables.
* Note:  Make sure to specify all locations with enough detail that
the dataset can be exactly reproduced.  So if a dataset includes
data from 196 counties in the US, please give explicitly the names
of all of the counties.
* Note:  Make sure to give the names of all the locations in a list
format.
* Example:  County vs State vs Country vs Census Block.
<notes> add a note if you have issues filling out any of the
fields.
Please copy directly the text of the paper.
Please be concise but have enough detail to reproduce.
Make sure you generate the JSON.

Here is the paper:
{{text}}
```

Figure 24: Prompt for extracting datasets from geospatial research papers (GEO Task).

To annotate the papers in the dataset, we worked with two annotators who carefully captured the dataset details for each paper in a standardized JSON format. If key information was missing, we asked the annotators to use domain expertise and common sense to make an educated guess, while also adding detailed information in the notes section highlighting the shortcomings of the paper. The work of the annotators was verified by experts with Ph.Ds. Agreeing on a reasonable format that correctly captured all details required to reproduce the work, while not being overly cumbersome to annotate or read, was challenging (e.g., sometimes it made sense to group variable names and time ranges, versus writing out the entire combinatorial set). After a few iterations, we arrived at a reasonable representation that we believe a human reader could understand clearly without ambiguity or error, while still being concise.

### F.5.3 GEO: DIFFICULTY RATING LOGIC

**Human evaluation consideration.** Extracting structured dataset information from unstructured scientific text poses unique challenges for evaluation, so we first present the components considered for evaluation prior to describing the logic for measuring difficulty. At the dataset level, we evaluated each model's ability to accurately predict the number of datasets mentioned and identify dataset names using text similarity metrics. At the variable level, evaluation focused on the model's accuracy in predicting the total number of variables, identifying individual variables based on a composite representation incorporating multiple attributes, and correctly extracting the time and spatial ranges of each variable.

Using the above rubrics, the difficulty of each example was rated as "easy" if the number datasets and the time and spatial ranges were more easily extracted from a single section of the paper. Examples were rated "hard" if the information was much more spread out and the annotators felt it was tricky to have captured all aspects accurately. Other examples were rated "medium".

### F.5.4 GEO TASK RESULTS ANALYSIS

Doing a full evaluation of all of the components described above allowed us to pay attention to the parts of the answers we cared about most, but to do this well one had to have similar formatting between the ground truth answer and the model response, and the models often got a lot of the right pieces of the answer but in a different format, or returned something that looked mostly correct but was not a parsable JSON, making programmatic metric calculations difficult. Other specific points about why this task was challenging to evaluate are: there were often multiple ways to format a correct answer, some parameters required exact matches (e.g., time range) and others semantic matches (like variable description), variable name and description were often muddled and salient information could be placed in either one, dataset name and variable name could sometimes be confused as well (sometimes it was reasonable to concatenate variable name, description, and dataset name and then do a semantic match on the whole set). The most success we had was with prompting an LLM to do the scoring using the LMScore, as described in Appendix D. We found that LMScore scores captured the salient features that mattered most automatically, hiding a lot of this complexity, and for this task the human evaluation results correlated quite strongly with the model-based scores.

A few overall observations: there was variability across model runs for almost all models, even though average performance remained fairly constant. The variability was usually higher on the harder tasks. Instruction following remains a challenge: models often had pieces of the right answer, but were unable to consistently format their responses, often leading to unparsable JSONs. The model-based scoring approach circumvented this challenge and also showed good correlation with human evaluations. ROUGE-L scores on the closed-weight LLMs were all quite similar (in range [28, 32]), with GPT-4o performing slightly higher than the others. Open-weight model scores were lower (in the 20's for Mixtral and Command-R+ and ¡ 10 for LongLLaMa). We explored a few different breakdowns of the task results to try to identify systematic correlations in model performance. The first breakdown, shown in Figure 26, plots score values versus the number of datasets in the paper, showing a slight inverse correlation between score and number of datasets, although the trend is very small and not very consistent.

We also analyzed performance as a function of geographical or spatial information. For studies encompassing one continent the performance was higher than for compound spatial ranges involving multiple continents (see Figure 27). Overall, results for this task illustrate that there is much room

Figure 25: Example from the GEO task where all closed models Gemini 1.5 pro, GPT-4o, and Claude-3 Opus extracted the correct spatial and time range for the datasets. They provided a less specific link than what human annotators provided and used a slightly different format where the models treated each variable as a separate dataset whereas human considered the dataset to be a single source but with different variables. Despite this slight difference the model's response is still accurate and useful. (paper id: 00000)

for improvement for LLMs to be able to automatically extract dataset information from the literature, especially in some of these more complex compound cases.

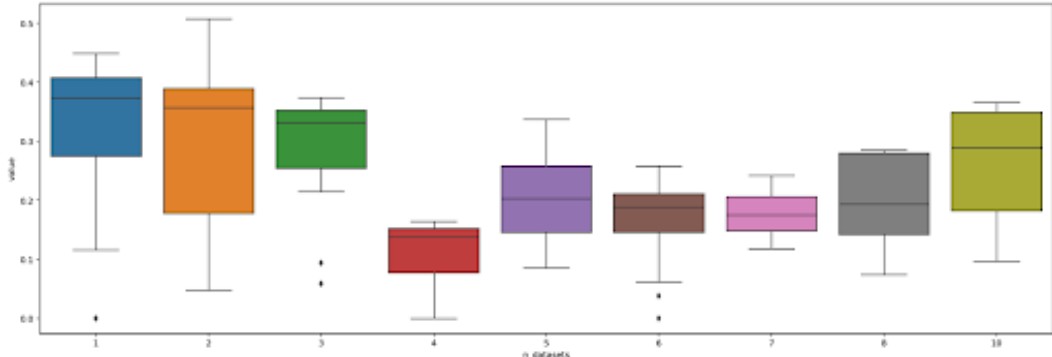

Figure 26: Score value versus number of datasets in the paper. There papers with fewer datasets have substantially higher performance, but there isn't a linear degradation in performance as the number of datasets studied in the paper increases perhaps because when many datasets are used in the paper they are still tied to some common sources.

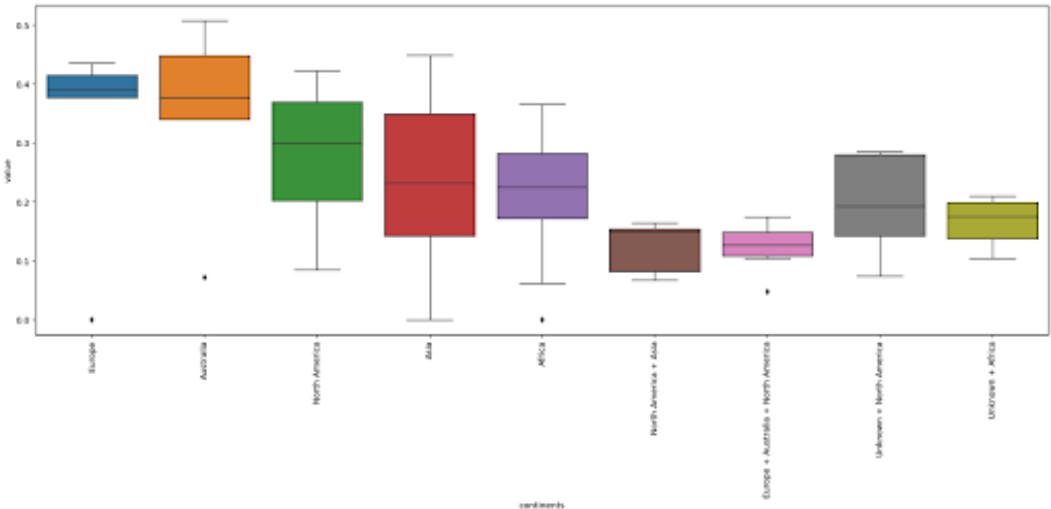

Figure 27: Score value versus geographical/spatial range studied in the paper. The scores for papers containing datasets covering multiple continents in their spatial extents (the 4 bars on the right) have noticeably lower performance compared to those covering the spatial extent within a single continent. Further, spatial ranges in Europe and Antarctica seem better scoped than other continents, this might be worth verifying with a large sampling of papers.

### F.6 BIODIVERSITY GEOREFERENCING TASK (BIOGR)

In this task we study the ability of multi-modal LLMs to work with geographical maps. Specifically, we investigate the core capability of georeferencing, where, given an image of a map and its associated caption, the task is to determine the latitude/longitude bounding box encompassing the region displayed. Figure 28 shows examples of images from the dataset georeferenced against geographical maps. This section provides additional details regarding the dataset creation, task definition, and evaluations for the biodiversity georeferencing task.

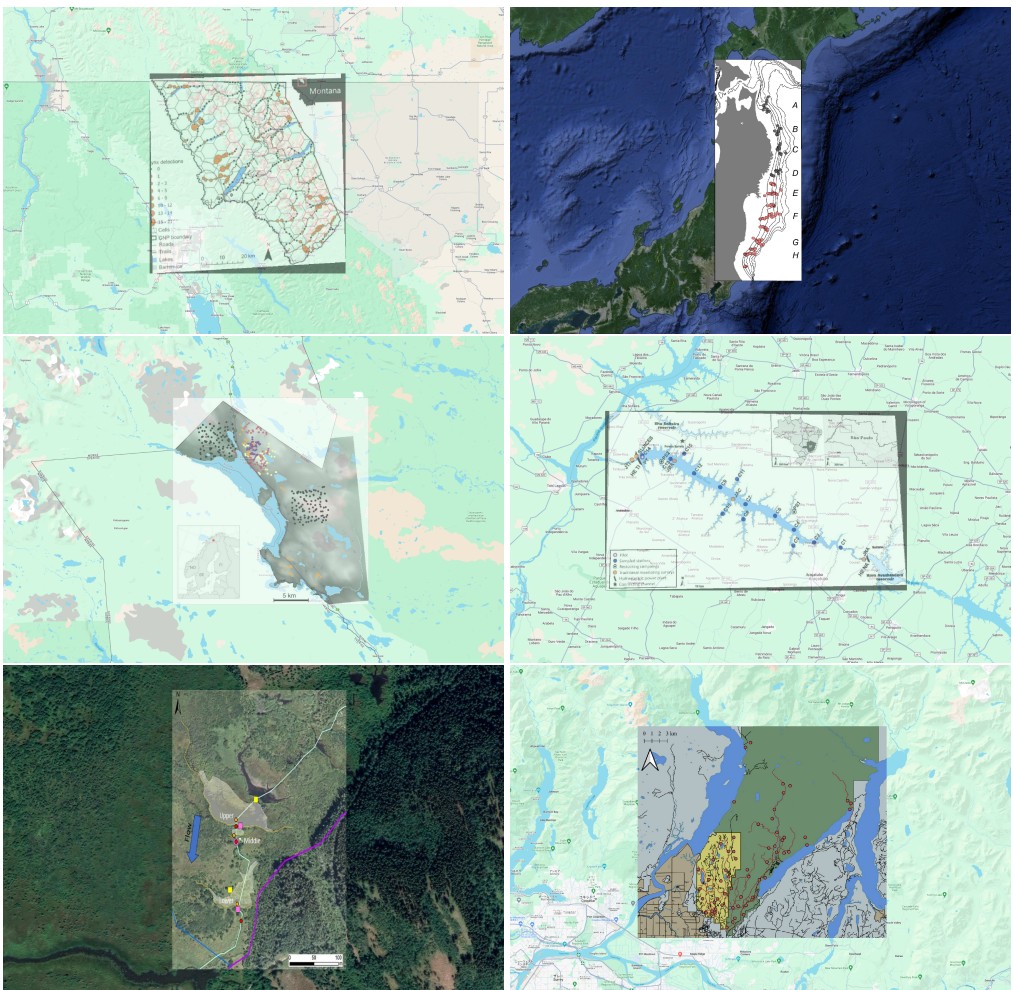

Figure 28: Illustrations of map georeferencing, showing images from the dataset superimposed over geographical maps in their correct georeferenced locations.

### F.6.1 BIOGR DATASET COLLECTION

To source this dataset we worked with domain experts in ecology to find articles that were representative of the domain, paying specific attention to the map figures. We pulled from various sources and repositories (e.g., bioRxiv, ScienceDirect, Nature, etc.), and included only articles with open licenses (e.g., CC BY 4.0 or CC BY-NC 4.0). We wanted to represent a variety of map features often displayed in these figures, as well as varying degrees of georeferencing difficulty (e.g., some regions are larger and some smaller, some mention locations explicitly in the image and/or caption, some have recognizable shapes at the continent or country level, some contain multiple repeated instances of the displayed region, some have insets or coordinates displayed to help the reader understand the spatial information, etc. Fig. 29 shows a few illustrative examples). We assembled a dataset of 24 map images and captions, extracted from 24 papers, and worked with a domain ex-

pert to acquire the ground truth latitude/longitude bounding box for each image. We also asked the domain expert to rate the individual tasks using a difficulty scale from 1-10 (1 being the easiest and 10 the hardest) which we simplify to the easy, medium, hard ratings as with other tasks. A portion of the BIOGR dataset consisting of 114 examples with additional metadata including original pdf documents from which the figures are sourced is available at https://github.com/google-research/ecology-georeferencing.

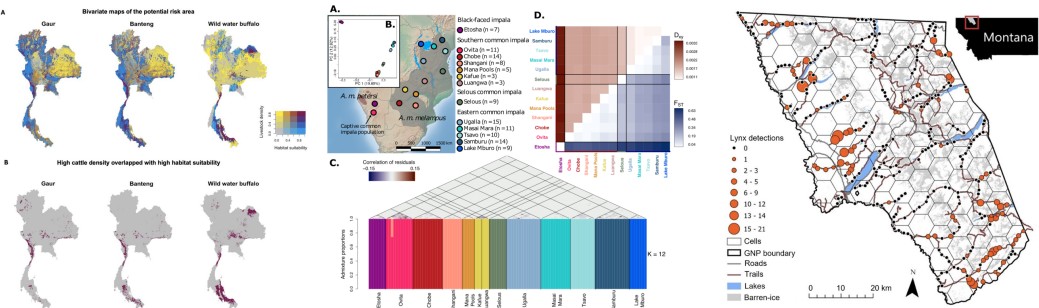

Figure 29: A sample of map images from the Biodiversity Georeferencing Task dataset.

### F.6.2 BIOGR TASK DETAILS

The goal for the BIOGR task is to accurate extract the spatial coordinates (latitude and longitude) for a location that is specified in the map images. To do this, we configured a standard prompt, and passed the prompt, along with each image and caption pair, to each of the multi-modal models evaluated. The prompt we used is shown in Fig. 30

---

**BIOGR prompt**

```
For the following figure and caption, please return the WGS84
latitude and longitude bounding box coordinates of the map in the
image.
If there are multiple maps of the same region, please just return
only one answer.
If two areas are represented, and one is an inset of the other,
return the smaller of the two areas.
If you are not sure, please guess at an answer anyway.  I'd rather
have an answer than no answer at all.
Make sure to return decimal coordinates in range [-90, 90] for
latitude and (-180, 180) for longitude.
Please put your answer in the following JSON format:
{
"W":  <west>,
"S":  <south>,
"E":  <east>,
"N":  <north>
}
Here is the image and caption.
{{text}}
```

---

Figure 30: Prompt for the Biodiversity Georeferencing Task (BIOGR).

Given this prompt, the image, and the caption, the selected model would provide a text response. Even though explicitly instructed not to, some models would provide a text preamble prior to the JSON bounding box answer. In these cases we filtered the response to extract the JSON bounding box answer. In all instances, all models were able to format their answer in the requested JSON format suggested in the prompt, so we did not have to handle cases where the answer was provided but not in the requested format. For illustration purposes, a sample ground truth answer looks like:

```
{
    "W":-114.4726175647,
    "S":48.2299910717,
    "E":-113.2291776519,
    "N":49.0025266467
}
```

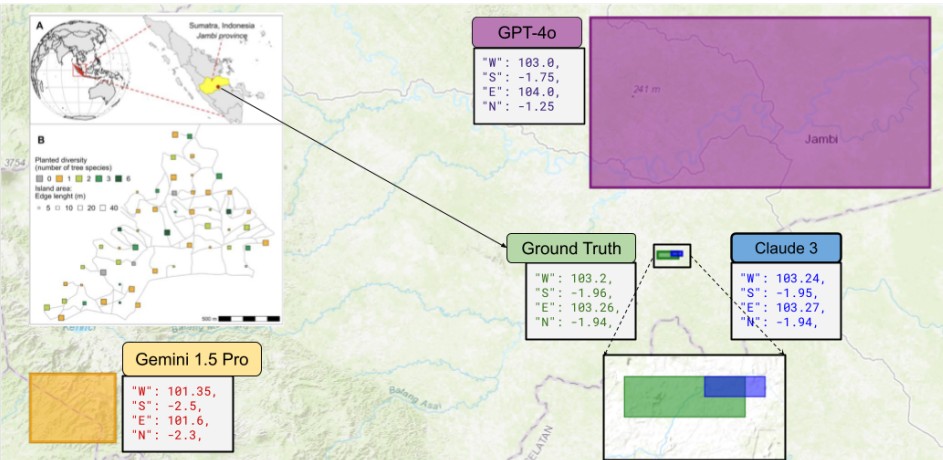

Figure 31: Responses from the different models for an example from the BIOGR task. Here, Claude-3 (blue) predicts the coordinates closest to the ground truth (green) bounding box locations indicated by the red dot in the original map image (top-left). GPT-4o (purple) and Claude-3 Opus generally performed well across examples on this task while Gemini 1.5 pro (orange) was a bit less precise. (record id: 556058_2)

### F.6.3 BIOGR RESULTS ANALYSIS

For this task we consider 3 evaluation metrics. **Intersection-over-Union (IoU)**, which is defined as the area of intersection between two bounding box regions divided by the area of union. IoU is a popular metric for bounding box tasks, and captures the similarity between two polygons, accounting for both location and size, while also being scale-invariant. For simplicity, we computed IoU using longitude and latitude as our X and Y axes directly versus converting into meters[4]. Explicitly, we used "W" and "E" as our minimum and maximum X-coordinates, and "S" and "N" as our minimum and maximum Y-coordinates respectively. **Normalized Distance Error** measures the distance between the centers of the two bounding box regions (in kilometers) when placed on the surface of the Earth, and this is normalized relative to the distance of the half-diagonal of the ground truth bounding box. This metric helps measure how far off the predicted box is when the prediction doesn't overlap with the ground truth box. **Relative box size.** This is the relative size of the predicted box (distance of the half-diagonal) with respect to the ground truth box half-diagonal, which is indicative of the error in the predicted box size.

When performing this task, over the full image/caption set and 3 model runs, the average IoU with each model ranged between 0.42 and 0.54 (Table 1), with GPT-4o performing slightly higher than the other models. The median results, with 25% and 75% error bars, are shown in Figure 5, with medians ranging between 0.3 and 0.6 and error bars showing wide variability accounting for both task and model variability. To put these numbers in context, the IoU metric ranges between 0 and 1, and any value greater than 0 denotes at least some region overlap, which implies that the model has at least some geospatial knowledge. Values between 0.2 and 0.5 have reasonable region overlap, and

---

[4]A more precise IoU calculation would convert to physical distance, which would equally penalize horizontal and vertical errors, and would account for the fact that the physical distance between two longitude lines varies with latitude; however, for small regions this difference is minor and the scale-invariant IoU metric captures the bounding box overlap reasonably well.

values greater than 0.8 have very good georeferencing performance (see Figure 32 for an illustration of IoU).

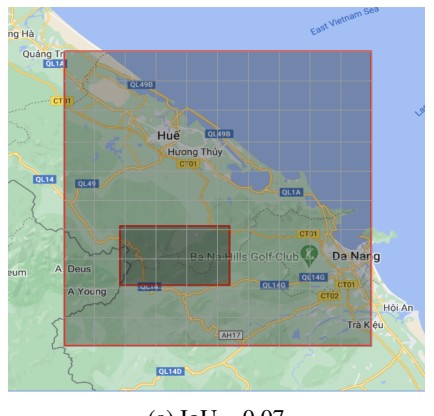
(a) IoU = 0.07

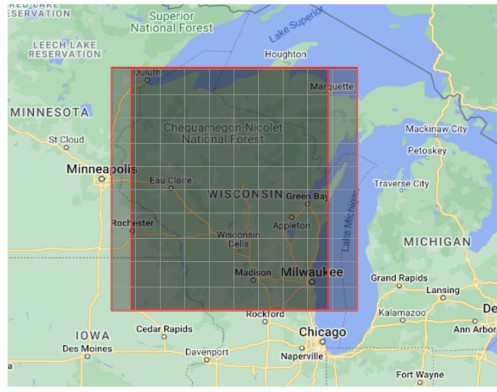
(b) IoU = 0.8

Figure 32: An illustration of Intersection-over-Union (IoU) metric for biodiversity georeferencing.

| Model | IoU ↑ | Normalized Distance Error ↓ | Relative Box Size ↓ |
|---|---|---|---|
| Claude 3 (Opus) | 0.39 | 6.15 | 1.39 |
| Gemini 1.5 Pro | 0.42 | 14.28 | 1.20 |
| Gemini 2.0 Flash | 0.49 | 3.03 | 3.05 |
| Gemini 1.5 Flash | 0.36 | 6.09 | 1.69 |
| GPT-4o | 0.37 | 1.43 | 1.22 |
| Gemini 1.0 Pro | 0.34 | 12.37 | 1.30 |

Table 7: **Performance of models on BIOGR** based on IoU, Normalized Distance Error, and Relative Box Size. ↑ indicates higher is better, ↓ indicates lower is better. Blue highlights the highest values across closed models.

One observation is that, even though we explicitly asked the models to provide answers even if they weren't sure, there were instances where models refused to estimate any coordinates. We also noted that there was some variability in model responses over the 3 runs for each image, and the variability was higher on the harder examples in the dataset. In additional experiments with Gemini 1.5 Pro, reducing the temperature of the model reduced variability but also increased the percentage of cases where the model refused to provide an answer. These stochastic and robustness performance issues must be taken into account when building systems that use LLMs.

To analyze how different features in the image/caption pair contributed to the LLM's ability to perform the task, we ran further experiments with Gemini 1.5 Pro, where we passed in only the image without the caption, and only the caption with no image. On the image only runs the average IoUs and model non-responses remained roughly the same, but on the caption only runs the IoUs decreased and the model non-responses increased significantly, highlighting that the image is much more important than the caption for this multimodal task. We also computed average IoUs over various subsets of the dataset, shown in Figure 33 on the left. The subset of images with coordinates shown on the map performed better than average, which is reasonable since multi-modal LLMs are trained on similar bounding box object detection tasks at the pixel level, and having a numerical scale present in the image that the model can reference increases performance. The larger regions were significantly easier for the model than smaller regions, partly because the chances of overlap are higher and partly because larger regions often had recognizable shapes or well-known locations mentioned. One result that was surprising was that, in this set of results, the images of small regions without insets performed better than the ones with insets, which is counter-intuitive. On closer inspection, the images in this set with small regions and without insets either had coordinates displayed, or were really well-known regions (e.g., maps of known cities), explaining the higher performance. Increasing the size of the dataset and adding more representation of images with insets and without coordinates or well-known shapes/locations would help generate more comprehensive

statistical results for the inset subset. Our suspicion is that these models are not currently paying a lot of attention to the image insets, like a human georeferencer would, and that there is room for improvement in teaching these models to process that information well. We plan on studying this further in future work. Finally, the right side of Figure 33 shows IoUs for all the tasks and models plotted against the domain expert's difficulty rating, suggesting that the tasks that were harder for the domain expert were also more challenging for the model (GPT-4o had the cleanest correlation trend, whereas the other models had more variability in their performance).

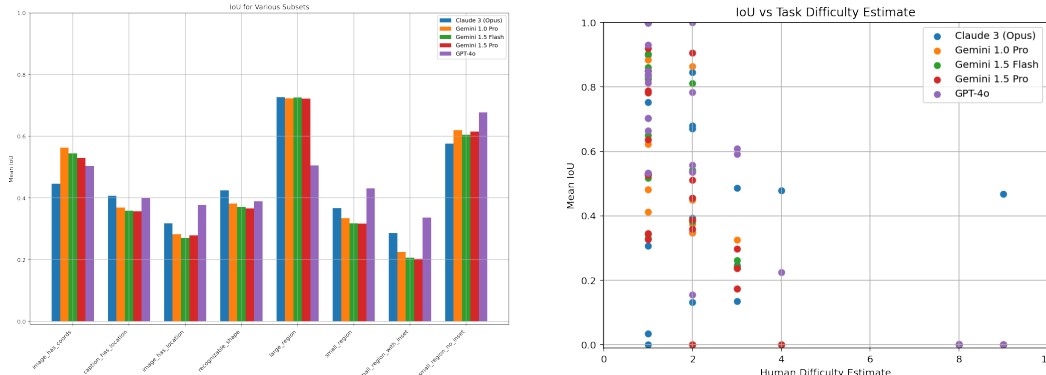

Figure 33: On a subset of 24 BIOGR examples, the left chart shows a breakdown of IoU for various data subsets. On the right we show IoU versus task difficulty ratings provided by a domain expert.

Finally, we note that we discarded some examples as out of scope when creating the dataset not because of difficulty, but because they did not facilitate clear query and a single accuracy metric (e.g. multiple unrelated images in the same figure, multiple study areas within a bigger region, cropped and concatenated regions that were not actually geographically contiguous, etc.). However, a system that automatically parses the literature to extract information will have to deal with these real-world examples as well. For future directions we would like to go deeper and build benchmarks that help study the ability of LLMs to extract more detailed information from maps (e.g. polygons of surveyed areas, points of interest highlighted in the legend, geographical features and insights that correlate to the data presented, Q&A pairs that capture what a domain expert would ask, etc.). Being able to work well with maps has wide applicability to many scientific domains (e.g. geospatial analysis, public health, epidemiology, etc.) so it's encouraging to see how much spatial information is already captured by these models today. One final note is that a domain expert usually uses a variety of tools at their disposal to perform these tasks (e.g., using ArcGIS, QGIS, Google Maps, etc.), rather than memorizing a large library of maps. It would be interesting to teach LLMs to integrate these geospatial tools as part of their workflows and analyze how much performance can increase via this agentic approach.

### F.7 PROTEIN SEQUENCE RECONSTRUCTION TASK (PDB)

This task assesses the ability of large language models (LLMs) to extract biologically relevant information from structured textual data. Specifically, we challenge the models to reconstruct the amino acid sequence of a protein given its three-dimensional structure in PDB format. The dataset consists of 64 PDB files representing diverse proteins with varying sizes, functions, and originating species. Each example comprises a single chain to maintain task focus and complexity.

Solving this problem necessitates an intricate understanding of the PDB format and the ability to parse it accurately. The model must first recognize the hierarchical organization of PDB files, identifying sections containing atomic coordinates and residue information. Subsequently, it needs to extract the three-letter amino acid codes for each residue, traversing the entire structure to ensure completeness. Finally, the model should leverage its knowledge of amino acid IUPAC Data to map the three-letter codes to their corresponding single-letter representations, ultimately producing a FASTA formatted sequence.

### F.7.1  PDB Dataset Collection

An expert in the domain identified 64 proteins from the Protein Data Bank[5] The proteins span a variety of species (15), sequence lengths, and protein functions. The ground truth amino acid sequences are also available in the data bank and can also be computed programmatically.

### F.7.2  PDB Task Details

For this task, we passed into the models PDB text describing the three-dimensional structure of a protein, along with a standard prompt asking the model to return the amino acid sequence in FASTA format. The prompt is shown in Fig. 34.

---

**PDB prompt**

```
You are a computational biologist and I want you to reconstruct a
protein's amino acid sequence from its tertiary structure.
* The input is a PDB that is a textual format describing the
three-dimensional structures of a protein.
* Return the amino acid sequence in the standard FASTA format,
which starts with a definition line with the greater than (>) line,
followed by the single-letter codes for all amino acids in the
second line.
* Make sure the amino acid sequence is in the second line.
* If there is an unknown amino acid in the structure, put "X" in
the sequence.
* Make sure you go through the whole structure and get all the
amino acids.
* No extra explanation is needed.

below are the tertiary structure:

{{text}}
```

Figure 34: Prompt for reconstructing protein amino acid sequences from tertiary structures (PDB Task).

### F.7.3  PDB Results Analysis

Evaluating the reconstructed sequences involves comparing them to ground truth sequences obtained using the Biopython library. We employ pairwise sequence alignment to determine the optimal correspondence between predicted and ground truth sequences. The alignment is then scored using standard metrics including the number of gaps, identities, and mismatches. These raw scores are normalized by the alignment length to account for potential length discrepancies. We primarily report the identity ratio, defined as the number of identities divided by the alignment length, as the primary performance metric. Unlike other tasks, model-based evaluation is not employed here due to the straightforward nature of exact evaluation and the limited ability of current LLMs to accurately assess sequence similarity. ROUGE score, typically used for text summarization, is also unsuitable for capturing the nuances of protein sequence similarity.

Our experiments revealed several failure modes, highlighting the challenges LLMs face in processing long and structured biological data. Frequently, the models struggled to accommodate the entire PDB file within their limited context windows. Even when the context window sufficed, the models often failed to traverse the complete structure, leading to incomplete sequences, particularly for larger proteins. Another recurring issue was the incorrect handling of consecutive identical amino acids, often resulting in undercounting. Additionally, we observed instances where the model mistakenly uses the first letter of the three-letter amino acid name instead of employing the established three-to-one letter mapping, despite this mapping being a fundamental concept in bioinformatics

---

[5]https://www.rcsb.org/

and likely encountered during the model's training. One noticeable trend was that the identity ratio dropped as the protein sequence became longer, which is reasonable, although we note that for the shorter sequence length proteins there was a large variability in performance (see Figure 35 for details).

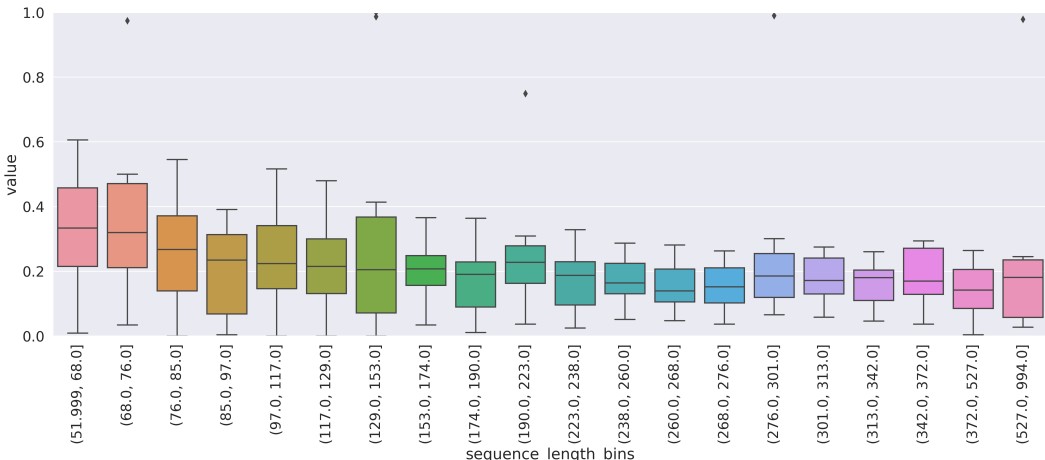

Figure 35: Plot of identity ratio versus protein sequence length, showing that performance drops as the protein sequence length increases.

