# OpenReview forum: "CURIE: Evaluating LLMs on Multitask Scientific Long-Context Understanding and Reasoning"
_ICLR.cc/2025/Conference — ICLR 2025 Poster_

### Official Review · Reviewer_CbxB · 2024-11-02

**Soundness:** 3
**Presentation:** 3
**Contribution:** 3
**Rating:** 6
**Confidence:** 4

**Summary:**

This paper introduces a new benchmark for long context understanding, reasoning and information extraction of different scientific tasks and evaluates the current state-of-the-art LLMs on it. It has 10 tasks split on 6 disciplines which are labeled by domain experts.
The authors also introduce a new LLM-based evaluation metric and use it along with evaluation established metrics. They also employed domain experts to evaluate the LLM predictions on a sample of the dataset.

**Strengths:**

- This work introduces a novel dataset with challenging tasks from different disciplines and evaluates different state-of-the-art LLMs and shows where they lack. I appreciate the effort that has went into building this dataset. I understand that this required time from domain-experts from different disciplines to label it. They have employed domain experts to make sure they have a reliable evaluation on the predictions of the dataset
- Having benchmarks for complex scientific domains is important for the improvement of language models
- The paper is well written and the message is conveyed clearly

**Weaknesses:**

- For some tasks there are not many examples
- As you mention, these answers are open-ended and it’s hard to evaluate the model’s output. A model may have a correct output but have used different set of steps or a different notation to answer. Having Latex as the ground-truth will also make it harder to evaluate. This will be less of an issue in the extraction tasks and more of an issue in tasks that require planning/reasoning. I understand that you use ROUGE-L/BERTScore F1, but I still think the performance may be unreliable for some of these tasks
- LLMSim/LMScore which follow LLM as a judge paradigm can also have their shortcomings. I would suggest adding multiple choices where it makes sense in this dataset. Even though you won’t be evaluating the expressiveness of the language model, it will be easier to judge the LLM’s knowledge using accuracy as a score which is interpretable
- Asking for the LLM to come up with an open-ended dictionary will make it harder to evaluate and map the generated dictionary keys to the ground truth keys as you do with LLMSim
- Since this dataset is cross-domain and requires such a specialized knowledge for each domain, it would be challenging to build a good model that performs consistently good across the different domains. Instead, a dataset that is focused on a specific domain and is more thorough would be better

**Questions:**

- Have you considered making more specific questions? For example, in Figure 3 the DFT example where you ask to identify all the input structures and this expects a big dictionary, can you make multiple questions instead for each key of the dictionary?
Would it make sense to provide only the keys of the dictionary and ask it to complete?
- When you say 434 examples curated from 273 research papers, what would an example entail?
- Since this benchmark is also about evaluating models in long contexts, a plot of the model performance as a function of the context length would be interesting
- Figure 18 needs bigger font size for the axis labels

---

> ### Author Response · Authors · 2024-11-22
>
> Thanks for your thorough review, detailed comments, and actionable suggestions. We like many of the ideas you have proposed in the weaknesses. We share our thoughts below on some that we might be able to address within the time frame for this conference.
>
> **Some tasks.. not many examples**
>
> (response also included in general comments)
>
> We agree it would be good to have more examples for some tasks to provide stronger statistical signal. In fact, we did have more papers for GEO and BIOGR domains but had to remove them closer to submission when we discovered they had licenses that prevented re-distribution of the papers. Since submission, *we have gathered several more annotations for the BIOGR task, and are working on adding more examples to the PDB tasks as well* to increase each to near 50 to provide a stronger statistical signal. We are still in the process of annotations and running evaluations. Just as a note of comparison, the now popular GPQA (https://arxiv.org/abs/2311.12022) dataset which tests expert level knowledge in biology, chemistry and physics contains 448 multiple choice questions; whereas our dataset covers disciplines requiring narrow expertise and has a wider range and complexity in responses which makes such a dataset more time and resource intensive to create. Thus, we believe the dataset and in particular annotations will have value despite the size.
>
>
> **Evaluation of open-ended responses, LLM judge short comings**
>
> We agree that evaluation of open-ended responses is challenging and you are also right that this is more of an issue where the task requires planning and reasoning. We note that for tasks where a structured output is preferred (e.g. GEO, DFT, MPV) our prompts provide the required JSON response format, and most LLM models – closed and open models – appear to adhere to instructions somewhat well to produce a JSON response. There are still definitely issues where the JSON formatting is not adhered to well.
> For our planning tasks, such as HFD, the prompts are framed more similar to summarization so ROUGE is somewhat appropriate and appears to reflect model performances.
>
> (As we note in the general response) Overall, for the retrieval tasks such as DFT-S, DFT-P, MPV, we recommend LLMSim since it is designed and better suited for these tasks exactly for the reasons you noted. For summarization and aggregation tasks such as HFE, HFD, QECC, and DFT-C, the existing ROUGE and BERTScore metrics are appropriate as is. We use LMScore as an additional model based evaluation to provide signal, and is not necessarily meant to replace other metrics just yet. For BIOGR, we  use Intersection Over Union; and for PDB we use ID_{ratio} which are both designed by domain experts and appropriate metrics well suited for the tasks.
>
> **Converting questions to multiple-choice**
>
> This is a great idea! We might be able to do this for some of our tasks where the eval is tricky, e.g. GEO, and perhaps some components of the HFD derivation tasks. The BIOGR and PDB tasks could also be converted to multiple choice more easily, but the metrics for them are already quite well defined, domain appropriate,  and programmable, so it might not be necessary. We are attempting to convert some of our GEO tasks to multiple choice variants, this will not be complete within the discussion phase, but if it works out we can include it in a final version. We really do like the idea, and will consider it for future versions as well.
>
> **Dataset focused on a specific domain**
>
> It is true that building good models that perform well across domains is difficult, and it is indeed appealing to develop more tasks focused in a specialized domain. In creating CURIE, our intention was to evaluate capabilities of models on a few domains requiring deep expertise, both to demonstrate the capabilities of LLMs in these areas and also highlight applications that can benefit from good general models to solve problems that are perceived as tedious by experts. We also agree that having datasets focused on a specialized domain would be more thorough and will include it as a future direction.
>
> Thanks again for your review and support of this work!

---

> ### Comment · Area_Chair_7fxT · 2024-11-25
> **Reminder: Rebuttal Deadline for ICLR 2025**
>
> Dear Reviewer CbxB,
>
> As the rebuttal deadline approaches, please kindly check the papers' discussion threads and respond to the authors' rebuttals. If you haven't had a chance to respond yet, I’d greatly appreciate your input soon. Your insights are invaluable to the authors and the review process.
>
> Thank you for your effort and support!
>
> Best regards,
>
> Area chair

---

> > ### Comment · Reviewer_CbxB · 2024-11-28
> >
> > Thanks for your responses, I will keep the current score. I think you are in the right path, but the dataset needs more refinement to be a good and reliable benchmark

---

### Official Review · Reviewer_dZb5 · 2024-11-03

**Soundness:** 3
**Presentation:** 4
**Contribution:** 4
**Rating:** 8
**Confidence:** 4

**Summary:**

The paper introduces CURIE, a benchmark for evaluating LLMs on scientific problem-solving that requires understanding long-context information and multi-step reasoning across six disciplines: materials science, condensed matter physics, quantum computing, geospatial analysis, biodiversity, and proteins. CURIE includes 434 tasks derived from 273 scientific papers, covering realistic scientific workflows such as information extraction, concept tracking, aggregation, algebraic manipulation, and multimodal understanding. This benchmark evaluates eight state-of-the-art models, revealing significant performance gaps, especially in tasks like protein sequencing. The paper aims to foster advances in LLM capabilities for scientific applications.

**Strengths:**

1. The paper proposed a new long-context scienceQA benchmark CURIE, containing 434 examples from 273 science literatures in different disciplines. The tasks are carefully collected and benchmarked with the help of domain experts.

2. The paper provided a detailed experimental analysis and many meaningful discussions about the challenges LLMs face in long-context scientific reasoning, especially regarding tasks like protein sequencing and geospatial data extraction, where complex, multi-step reasoning and domain-specific knowledge are crucial. The appendix also included many details for benchmark construction, examples, and case studies, helping readers better understand the annotation process and challenges of the tasks.

3. The paper is well-written and easy to follow. I really appreciate the experts' efforts for annotation and the authors' contribution for conducting this research, and believe this benchmark would be beneficial for both developing better LLMs and better usage of LLM for scientific research.

**Weaknesses:**

1. I'm not sure if LLMs could have seen some of these research papers, which might affect the effectiveness of evaluation results on CURIE. A figure showing the comparison / correlation of performance between CURIE and other scientificQA / long-context QA benchmark would resolve this concern more clearly.

2. I think one of the challenge of such benchmarks should be the difficulty of scaling up? Due to the static manner and limited scale, public papers might be included within future (or existing) corpus for LLM training. I wonder if the authors considered to somehow automatically scale up the dataset, or keep updating the examples?

**Questions:**

See "weakness". I've listed my questions there.

---

> ### Author Response · Authors · 2024-11-22
>
> Thanks for your accurate summary and endorsement of this work, we appreciate your comments and suggestions to further enhance this work.
>
> **LLM familiarity with papers and correlation of performance on scientific QA and long-context benchmarks**
>
> This is a nice suggestion to compare performance across different benchmarks broadly related to knowledge in papers vs. a more specific problem-solving benchmark on the papers. Figure 1 (b) provides some comparison with GPQA, DROP and MMLU. We will compare this with some long context benchmarks as well and add a figure to observe correlations.
>
> It is very likely that all the models have seen many (if not all) of the papers in our benchmark. Our papers are sourced from arxiv and previous corpora such as the S2ORC corpus (https://github.com/allenai/s2orc). Though, it is unlikely that the models have encountered similar tasks during their training or tuning stages, especially on such scientific literature. Another, key difference is that our tasks do require some degree of planning and organizing, so simple agentic frameworks like ReACT might do slightly better but they need to be written specifically for each task by experts, perhaps tuned to some extent for each paper.
>
> **Scaling up**
>
> (also included in general response)
>
> We agree such dataset creation requires human (and actually expert) annotations and scaling this is a challenge. Though related to your first comment on papers being included in the pre-training may be fine. For some tasks, such as PDB, it might be easy to scale up the dataset and evaluation since there is a programmatic way to generate a solution. However, if the input and output pairs are used in model training then it would render the evaluation somewhat less useful. With newer agentic frameworks developing, we will think more about how examples may be automatically updated and verified. We can include this as a future direction.
>
> We hope to add the figure to observe correlations with other science QA and long context tasks to the updated version of the draft soon. Thanks again for the suggestion!

---

> > ### Comment · Reviewer_dZb5 · 2024-11-28
> > **Official Comment by Reviewer dZb5**
> >
> > Thanks to the authors for your thoughtful and thorough response. Having reviewed the authors' response as well as the other reviews, I stand by my scores for this work.

---

> ### Comment · Area_Chair_7fxT · 2024-11-25
> **Reminder: Rebuttal Deadline for ICLR 2025**
>
> Dear Reviewer dZb5,
>
> As the rebuttal deadline approaches, please kindly check the papers' discussion threads and respond to the authors' rebuttals. If you haven't had a chance to respond yet, I’d greatly appreciate your input soon. Your insights are invaluable to the authors and the review process.
>
> Thank you for your effort and support!
>
> Best regards,
>
> Area chair

---

### Official Review · Reviewer_zECm · 2024-11-04

**Soundness:** 3
**Presentation:** 3
**Contribution:** 2
**Rating:** 6
**Confidence:** 4

**Summary:**

This paper introduces a benchmark called Curie for evaluating Large Language Models (LLMs) on science-related tasks in the disciplines of material science, condensed matter physics, quantum computing, geospatial analysis, biodiversity, and proteins. The benchmark includes 434 examples taken from 273 research papers related to the previously mentioned scientific disciplines. Based on those examples, tasks were created and used to evaluate several LLMs.

**Strengths:**

- The paper is well-written.
- The paper provides a helpful benchmark for evaluating LLMs in science-related tasks.

**Weaknesses:**

- There is no comparison with the baseline case of just inserting the task description into the LLM prompts without providing the paper. It would be useful to see what the results would be in that case.
- Only 6 very specific disciplines (material science, condensed matter physics, quantum computing, geospatial analysis, biodiversity, and proteins) are considered in this dataset. This limits the scientific domain in which LLMs can be tested.
- The dataset construction requires human annotations. This makes scaling up the dataset challenging.
- Typo line 339: "precison" -> "precision"

**Questions:**

- Have the authors considered inserting the task description into the LLM prompts without providing the paper? How would that compare with the CURIE method?
- Have the authors considered adding more tasks from a wider range of disciplines as future work?

---

> ### Author Response · Authors · 2024-11-22
>
> Dear reviewer zECm,
>
> ### Thanks for your time and effort reviewing this work and for sharing your comments. Based on the questions and weaknesses we believe that *there may be some misunderstanding of the paper/proposed benchmark*.
>
> **Baseline comparison without the paper**
>
> While this is a nice suggestion, it might not apply directly to many of our tasks. We would like to clarify that our benchmark is an active problem solving benchmark that requires understanding of the long context (Abstract Lines 011-015, Introduction Lines 035-038). Unlike MMLU or GPQA benchmarks that test “knowledge” of the LLM by asking different questions about the domain, our benchmark focuses on a specific problem that LLMs can help solve in each domain. As noted by examples in Figure 3, the task and subtask in each domain is fixed, the input context or paper is the one that changes with each question.
> So is the request to just evaluate the response where there is no paper, so we just have 1 response for each task and we can see what the baseline output looks like. It would help if you can clarify more.
>
> Section 2, Related Works (Lines 085-094) provides several examples of NLP tasks such as relation extraction, entity recognition, etc. in the science domain that in previous works operate on a single sentence or passage, whereas our benchmark attempts to perform similar tasks at the full-paper level.
>
> For additional clarification, Section 2, Related Works, Lines 101-107, describes other long-context benchmarks in a similar vein to ours (e.g. ZeroScrolls, NIAH, RULER etc.) and notes some of the tasks that these benchmarks contain like summarization, textual entailment, translation etc. But these are not in the scientific domain as noted in the Table in Figure 2 (a), whereas our tasks are all in scientific domains, and utilize scientific literature and papers as context as opposed to news articles or story books etc. used by general long context benchmarks.
>
> **Only 6 disciplines chosen**
>
> We believe this may also be stemming from some misunderstanding of the dataset. While we choose 6 disciplines to evaluate the LLMs on, the tasks and sub-tasks themselves could potentially apply to many other domains, but one would also need experts in those domains to create annotations. E.g. If we consider the MPV task of extracting materials, values and properties, it is a task that combines “named entity recognition”, “relation extraction”, “information extraction”, and “grounding”  in the traditional sense. All of these need to be done in the single MPV task. But such a combination is not unique to the material science domain, and similar tasks are applicable in other domains e.g. biomedical domain for PICOs extraction (Lines 089-091).
> Our main contribution is identifying and defining such tasks as they pertain to a specific domain, and creating annotations and evaluations for these tasks on those domains.
>
> Our hope is that this dataset would inspire scientists in other domains to also create such tasks as it pertains to their domain and area of expertise.
>
> **Requires human annotations, challenges in scaling up**
>
> (also included in general response)
>
> We agree such dataset creation requires human (and actually expert) annotations and scaling this is a challenge. CURIE is a new benchmark that fills in the gap in existing literature to curate a range of problem solving tasks in science. For instance, early works such as [1] that aimed to look at the impact of LLMs for science were a collection of anecdotal examples noted by experts. In contrast, our work aims to provide a concrete measurable benchmark. Similarly, the very recent FrontierMath [2] benchmark was also created with the help of expert mathematicians and professors coming together, and required human annotation. As we progress, we believe we might be able to identify datasets and tasks from experts in different scientific fields that might enable easier scaling. For instance, the PDB task in our dataset is quite amenable to scaling.
>
> [1] The Impact of Large Language Models on Scientific Discovery: a Preliminary Study using GPT-4. https://arxiv.org/abs/2311.07361
> [2] FrontierMath: A Benchmark for Evaluating Advanced Mathematical Reasoning in AI
> https://arxiv.org/abs/2411.04872
>
> **More tasks from more disciplines as future work**
>
> **Yes**, indeed. We are working with more experts in different fields and hope to add additional evaluations either as part of this benchmark or separately. In particular, an interesting line of work is to do it in scientific fields that use computation like [3]. This is a future direction we hope to pursue.
>
> [3] SciCode: A Research Coding Benchmark Curated by Scientists.  https://scicode-bench.github.io/
>
>
>
> ### We hope we have clarified any misconceptions of this dataset, and look forward to addressing any other concerns in the remainder of the discussion period.

---

> ### Comment · Area_Chair_7fxT · 2024-11-25
> **Reminder: Rebuttal Deadline for ICLR 2025**
>
> Dear Reviewer zECm,
>
> As the rebuttal deadline approaches, please kindly check the papers' discussion threads and respond to the authors' rebuttals. If you haven't had a chance to respond yet, I’d greatly appreciate your input soon. Your insights are invaluable to the authors and the review process.
>
> Thank you for your effort and support!
>
> Best regards,
>
> Area chair

---

> > ### Author Response · Authors · 2024-11-26
> > **Request to acknowledge response**
> >
> > Dear reviewer zECm,
> >
> > We have added additional tables to clarify the tasks (in response to reviewers rvRN and ECw3) and have included these along with figures showing input prompts for each task, and plots comparing correlation of performance on this benchmark vs other scientific QA and long-context benchmarks (Appendix B). Hope these help clarify the tasks and the contributions of this paper better.
> >
> > We would appreciate it if you can acknowledge the response and let us know if they address your concerns and will consider revising the score or if there are other points we can help clarify.

---

> ### Author Response · Authors · 2024-11-28
>
> Dear reviewer zECm,
>
> We would appreciate it if you are able to take a look at our responses, additional details added to the paper, and let us know if they address your concerns. We would greatly appreciate your reconsideration based on our additions in response to your valuable feedback. Thank you for your time and careful consideration of our work.
>
> Thanks,

---

> ### Comment · Reviewer_zECm · 2024-12-02
>
> Thank you for your responses. I have raised my score.

---

### Official Review · Reviewer_EWc3 · 2024-11-04

**Soundness:** 3
**Presentation:** 3
**Contribution:** 3
**Rating:** 6
**Confidence:** 3

**Summary:**

The paper introduces the CURIE benchmark, designed to evaluate large language models (LLMs) on multitask scientific problem-solving, focusing on long-context understanding and reasoning. CURIE encompasses ten challenging tasks across six scientific disciplines. These tasks require comprehension of extensive context and domain expertise.

The authors assess various LLMs, highlighting that while Claude-3 performs well across domains, models like GPT-4o struggle with specific tasks, such as protein sequencing. The study finds that current models still have significant room for improvement.

**Strengths:**

- The proposed benchmark focuses on long-context understanding and reasoning in scientific domains, which is relatively unexplored and meaningful for advancing AI4science.
- The paper provides many details and discussion towards the tasks and experiments.

**Weaknesses:**

1. The presentation of this paper is not clear enough. While the authors spent much space introducing the domains and tasks, the input and output (e.g., format, content) are not explicitly defined, which makes it difficult to understand how the authors evaluate the models.
2. The evaluation metrics may not fairly show the performance. The reliance on ROUGE-L and BERTScore may not fully capture the complexity of scientific reasoning. While the authors propose metrics like LLMSim and LMScore, how they extract the answers from open-formatted model outputs and compare the potentially heterogenous answers (e.g., different field name, spelling difference, number/string, with/without extra text) with the gold standard is not clear to me. If not carefully handled, the experimental results might not be reliable.
3. The experiment analysis does not provide many deep and insightful observations.

**Questions:**

1. Could the authors explain more about how you define the input and output format?
2. What is your method to evaluate the free-form generation?
3. What are the approaches for quality control?

---

> ### Author Response · Authors · 2024-11-22
> **Response 1-of-2**
>
> Thanks for taking time to review our work.
>
> **Presentation … not clear enough (weakness 1) and Question 1 - input / output format**
>
> Our motivation for describing the tasks is to help researchers understand potential applications of the models, and implication to the domain. We provide the prompts, the ground truth, and code to run inference in the supplement zip file.
> > We will also include input prompts for tasks, and add it to the appendix section of each task in the draft.
>
> The ground truth examples are in json format. Since the actual ground truth examples are rather long, Figure 3 shows portions of the ground truth outputs for 4 tasks.  Partial ground truth samples and model responses are provided for different tasks in Figures 6, 8, 14, 16, 17, 20, 21, 23, 25,and 31 (figure numbers correspond to updated draft). The ground truth outputs in some cases are rendered to improve visualization e.g. Figures 20 and 21 render latex equations, and Figure 31 renders the bounding box area, but the model outputs are text.
>
> The full prompts, ground truth, inference code and evaluation code have been included in the zip file with the supplement. The model responses from all models exceeded the memory limitation of the allowed supplement zip but will be included in the version of the data we release.
>
> **(weakness 2, question 2) Evaluation metrics, extracting answers from open formatted model outputs**
>
> (also in general response)
>
> We understand this concern, we have included the evaluation code in the supplement zip file. The advantage of ROUGE-L, BERTScore and LMScore metrics are that they do not require parsing of model output text.
>
> The LLMSim metric is used for the retrieval tasks and expects model output text to roughly conform to json format, we incorporate several heuristics to give additional leeway when model responses do not conform to this format and do a best effort match between ground truth elements and model responses by making several calls to the LLM. Specifically given a list of ground truth dictionaries and predicted list of dictionaries, LLMSim uses an LLM rater to verify similarity of each field/(key, value) in a dictionary and evaluate several components of the response to try to establish a 1-to-1 mapping between predictions and ground truth.  After the mapping is made, we compute precision and recall of predicted dictionary items with respect to the ground truth dictionary elements.
>
>
> (As noted in the general response) Overall, for the retrieval tasks such as DFT-S, DFT-P, MPV, we recommend LLMSim since it is designed to suit these tasks. For summarization and aggregation tasks such as HFE, HFD, QECC, and DFT-C, the existing ROUGE and BERTScore metrics are appropriate as is. We use LMScore as an additional model based evaluation to provide signal, and is not necessarily meant to replace other metrics just yet. For BIOGR, we  use Intersection Over Union; and for PDB we use ID_{ratio} which are both designed by domain experts and appropriate metrics well suited for the tasks.
>
> **Experiment analysis, insightful observations**
>
> We conduct several analysis of model responses and report overall summary across groups of tasks in the main paper, for example:
>
> (1) Section 5, Lines 412-430 and Table 2 report human vs model-based eval comparisons on retrieval tasks in material science (DFT-S, DFT-P and MPV tasks).
>
> (2) Section 5, Lines 439-466 including Figure 7 report performance of models on examples of different difficulty levels “easy”, “medium” and “hard”.
>
> (3) In the Appendix Table 4, we compare ROUGE-L with proposed model based metrics LMScore and LLMSim.
>
> (4) Additional analysis are also included in the appendix e.g.
>
>    (4.1) Appendix Figure 17, and Figure 18 (updated draft Fig. 26, 27) for the GEO task shows the variation in scores based on the number of datasets used in a paper, and the extent of the geographical/spatial ranges studied in a paper.
>
>   (4.2) Appendix Figure 24 (updated draft Fig. 35) shows how performance on the PDB task drops with increasing length of the amino acid sequence.

---

> ### Author Response · Authors · 2024-11-22
> **Response 2-of-2**
>
> **Regarding insights**
>
> (1) Sec. 6, Lines 476-481 note that all models struggle to exhaustively retrieve all elements from long context tasks, i.e. multiple-needles-in-a-haystack problem in the science context  for tasks such as MPV, and GEO, but are better on single-needle-in-a-haystack problem such as HFE.
>
> (2) Sec. 6, Lines 482-515 and Figure 8 show why models such as Claude-3 outperform other models on some tasks, specifically Claude-3 is able to understand the purpose of DFT calculations better and hence groups results in a manner that is much more well suited for the task.
>
> (3) Figure 6 also shows why models such as GPT-4o fail on the PDB task where model response generation seems to consistently be derailed into a repetitive cycle reminiscent of typical early recurrent language model works.
>
> (4) Sec.6, Lines 485-511 (QECC analysis) and also Appendix C.2.4 (MPV tasks analysis) note that models are good at combing out lot of details sometimes too trivial, some times incorrect, but experts find this behavior more tolerable since they find it easier to discard the trivial information but harder to consolidate components from different parts of the text.
>
> Other insights from expert annotators of each task are included in the appendix, we share some below and will try to highlight these better in the main paper or consolidate them in the appendix
>
> (5) Appendix Sec. C1.3. (revised draft F1.3) for DFT notes that some models such as GPT-4o which have a predisposition to brevity often fail to retrieve all sets of parameters for DFT-P subtask.
>
> (6) Appendix Sec. C6.3 (revised draft F7.3) for PDB notes observations where models mistakenly use the first letter of the three-letter amino acid name instead of employing the established three-to-one letter mapping, despite this mapping being a fundamental concept in bioinformatics.
>
> There are many more such analysis and insights from experts in the appendix: C.2.4 (revised draft F.2.4) for MPV; C.2.9 (revised draft F.3.4) for HFD/HFE tasks; C3.3 (revised draft F.4.3) for QECC; C.4.4 (revised draft F.5.4) for GEO; C5.3 (revised draft F.6.3) for BIOGR, and  C6.3 (revised draft F.7.3) for PDB tasks.
>
> > We can consolidate this and add an additional section in the appendix for overall observations and insights.
>
> **Quality control, question 3**
>
> Quality of the annotations comes from several aspects of the dataset design and annotation process.
> 1. Firstly, domain experts themselves identified and suggested tasks that they use in their workflows that they would like assistance on. So in many cases, our domain expert annotators already have expertise in the tasks and have created data of similar nature in other contexts.
>
> 2. Second, to establish consistent output formats and aid with evaluations, we had pilot annotation phases on many tasks notably, MPV, DFT, HFE, GEO, BIOGR etc. where we perform an initial set of annotations, and request experts to examine each others’ output and have an adjudication discussion to establish the output formats.
>
> 3. Other tasks like QECC, and PDB are already based on similarly curated data, but the task and output formats are modified by the experts to suit our overall goal for a long-context evaluation benchmark. Here we focus the task and response on the given input context, and establish the appropriate ground truth response and format.
>
> 4. Annotations for such tasks is a resource intensive process and annotators were compensated for their expertise, time, annotations, and help with identifying appropriate evaluations to ensure overall quality of the outputs.
> We firmly believe that the quality of the ground truth is very high. Beyond this benchmark, we hope that the rich human annotations can serve the community in advancing planning, instruction following, and also better evaluation techniques for generated texts of mixed and heterogeneous formats including dates, locations, numerical values, units, descriptors, domain specific terms, equations and code.
>
> ### We hope that we have addressed your concerns, and that you will consider revising your overall rating of this work. We are happy to address any further questions.

---

> > ### Comment · Reviewer_EWc3 · 2024-11-24
> >
> > Thank you to the authors for their detailed response and updated draft, which improved the clarity and addressed my concerns. I believe it would be beneficial to consolidate the tasks (e.g., their intro, task types, required abilities, evaluation protocols, metrics, more than just Table 3) and the insights, which may make it clearer and more attractive for people to use the benchmark.

---

> > > ### Author Response · Authors · 2024-11-25
> > >
> > > Thank you! This is a really good idea and would add more clarity to the evaluations as well. We'll add this table.

---

> > > > ### Author Response · Authors · 2024-11-26
> > > >
> > > > We have now added this table as Appendix Table 4 in the paper. We will try to work on consolidating Tables 3 and 4 to fit into a single table in the pdf (or at the very least when we release the benchmark)
> > > >
> > > > | Task | Brief description | Capability | Output Format | Metrics (Programmatic) | Metrics (LLM-based) |
> > > > |---|---|---|---|---|---|
> > > > | DFT-S | Extracts input material structures for DFT calculations. | entity recognition, concept tracking | JSON | ROUGE-L | LLMSim, LMScore |
> > > > | DFT-P | Extract parameters for DFT calculations. | concept extraction, tracking, aggregation | JSON | ROUGE-L | LLMSim, LMScore |
> > > > | DFT-C | Write functional code for DFT computations. | concept aggregation, coding | TEXT | ROUGE-L | LMScore |
> > > > | MPV | Identify all instances of materials, their properties, and descriptors. | entity recognition, concept extraction, tracking | JSON | ROUGE-L | LLMSim, LMScore |
> > > > | HFD | Derive the Hartree-Fock mean-field Hamiltonian for a quantum many-body system. | concept extraction, algebraic manipulation, reasoning | TEXT | ROUGE-L | LMScore |
> > > > | HFE | Extract the most general mean-field Hamiltonian. | concept extraction | TEXT (latex equation) | ROUGE-L | LMScore |  |  |
> > > > | QECC | Create a YAML file with the Error Correction Code’s properties. | concept aggregation, summarization | YAML | ROUGE-L | LMScore |
> > > > | GEO | Extract information for all geospatial datasets used along with the spatial and temporal extents. | concept extraction, aggregation | JSON | ROUGE-L | LMScore |
> > > > | BIOGR | Determine the latitude, longitude bounding box encompassing the region in the map image. | visual comprehension, reasoning | JSON (lat., lon. co-ordinates) | Intersection-over-Union (IoU) | -  |
> > > > | PDB | Reconstruct a protein’s amino acid sequence form the 3D structure. | reasoning | TEXT (amino acid seq.) | Identity ratio (IDr) | -  |

---

> > > > > ### Comment · Reviewer_EWc3 · 2024-12-01
> > > > >
> > > > > Thanks for the update!

---

### Official Review · Reviewer_rvRN · 2024-11-04

**Soundness:** 2
**Presentation:** 2
**Contribution:** 3
**Rating:** 6
**Confidence:** 3

**Summary:**

This paper introduces CURIE, a scientific long-context reasoning and information retrieval (IR) benchmark. It contains ten tasks across six scientific domains, all designed to be challenging yet realistic. A wide range of open-source and closed-source large language models (LLMs) are tested, and Claude-3 consistently outperforms other models, including GPT-4.

**Strengths:**

- **Dataset Design**: The dataset is well-designed, covering ten complex tasks (even complex for humans) across six scientific fields. It targets the realistic problems faced by scientists.

- **Annotation Process**: The annotation process is thoroughly explained, and the motivation for selecting each sub-task is clearly articulated.

**Weaknesses:**

**Dataset Size Too Small**:

- “The CURIE benchmark encompasses 434 examples across ten tasks curated from 273 research papers across six diverse scientific disciplines.” The scale of the dataset is somewhat small. Specifically, tasks like PDB and CEO have only 21 and 19 examples, respectively. This limited size may lack statistical significance when comparing models at the sub-task level, making it difficult to determine the reliability of the results given the small sample sizes.

**Issues with LMScore**:

- As mentioned in Appendix A, GPT-4 is used as the language model (LM) to compute LMScore. It would be better to use an open-source model, as GPT models are constantly being updated, which may alter evaluation scores over time. Even when specifying a GPT version, it may become deprecated, making it hard to replicate the results.

- Additionally, in Figure 9, the results do not convincingly show that LMScore has a high correlation with human judgment. Therefore, it may be challenging to conclude that LMScore can replace ROUGE.

**Presentation Could Be Clearer**:

- It would be beneficial to include a table that specifically lists the number of questions, the number of questions under each task, and the average number of tokens (or words) in queries, documents, and ground truths. Although this information is scattered across Figure 2(b)(c) and Figure 4(c), consolidating it into a table would enhance clarity.

- Line 269: “On the extraction tasks, such as MPV, HFE, GEO, and DFT-S, experts within each domain reviewed each other’s work and reported a high rate of agreement.” It would be better to provide statistics on the inter-agreement between annotators.

**Questions:**

- I don’t fully understand how LLMsim works. If the goal is to identify the number of dictionaries that are correctly retrieved, why not use an exact match or other metrics commonly used for evaluating retrieval, such as nDCG?

- For each paper, is there only one question asked? The paper is not entirely clear on this point, and it raises the question of whether each paper corresponds to only one question or several questions (similar to Qasper).

- In Figure 4(c), does the average length of 954 words refer to the ground truth, or does it refer to the average length of the model-generated responses?

---

> ### Author Response · Authors · 2024-11-22
> **Response 1-of-2**
>
> Thanks for your nice summary, positive feedback on the dataset design and annotations and thoughtful comments on our work.
>
> **Dataset size** On the dataset size, we agree that this may pose a limitation on statistical significance in some of the subdomains. We in fact did have more papers for GEO and BIOGR domains but had to remove them closer to submission when we discovered they had licenses that prevented re-distribution of the papers.
>
> Since submission, we have gathered several more annotations for the BIOGR task, and are working to add more examples to the PDB tasks as well to increase each to near 50 to provide a stronger statistical signal. We are still in the process of annotations and running evaluations. Just as a note of comparison, the now popular GPQA (https://arxiv.org/abs/2311.12022) dataset which tests expert level knowledge in biology, chemistry and physics contains 448 multiple choice questions; whereas our dataset covers disciplines requiring narrow expertise and has a wider range and complexity in responses which makes such a dataset more time and resource intensive to create. Thus, we believe the dataset will have value despite the size.
>
> **Open source model for LMScore** This is a good idea! We are working on adding an evaluation with open source models and hope to have something in time during this discussion phase.
>
> **LMScore, human judgements, and LLMSim**  The key challenge in our evaluation lies in identifying 1-to-1 mapping from generated language text to ground truth elements since the correct answer needs to encompass many components. This is the reason we propose LLMSim as the metric in the main paper. While LMScore tries to compare the ground truth response and predicted answer in a single LLM call, the LLMSim metric makes many more calls to the LLM trying to verify similarity of each field/(key, value) in a dictionary and evaluate several components of the response trying to accurately gauge both precision and recall of predicted values and the ground truth elements. As noted in the paper we use LMScore as an additional model based evaluation to provide signal, and is not necessarily meant to replace other metrics just yet. For the retrieval tasks such as DFT-S, DFT-P, MPV, we recommend LLMSim since it is better suited. For summarization and aggregation tasks such as HFE, HFD, QECC, and DFT-C, the existing ROUGE and BERTScore metrics are appropriate as is. For BIOGR, we  use Intersection Over Union, and ID_{ratio} for PDB which are both domain appropriate metrics well suited for the task.
>
> **Presentation, Table** Agreed. We have consolidated the information in a table and will include it
>
>
> | Domain                  | Task | # papers | # examples | Avg. input length | Avg. ground truth output length |
> | ------------------------ | ------- | ---------------- | ------------------- | ----------------- | ------------------ |
> | Material Science         | DFT-S   | 74               | 74                  | 5818              | 232                |
> | Material Science         | DFT-P   | 74               | 74                  | 5818              | 132                |
> | Material Science         | DFT-C   | 74               | 74                  | 5818              | 1742               |
> | Material Science         | MPV     | 17               | 17                  | 1687              | 2188               |
> | Protein Sequencing       | PDB     | 21               | 21                  | 44028             | 12                 |
> | Geospecial               | GEO     | 19               | 19                  | 7802              | 808                |
> | Quantum Computing        | QECC    | 65               | 65                  | 19913             | 207                |
> | Condensed Matter Physics | HFD     | 15               | 15                  | 5385              | 1422               |
> | Condensed Matter Physics | HFE     | 38               | 38                  | 8472              | 111                |
> | Biodiversity             | BIOGR   | 24               | 24                  | \--               | 20                 |
>
> **Presentation, Inter-annotator agreement**
> We will include the exact numbers for annotator agreement.
>
> * For HFE and DFT-S task the agreement was near perfect (>95%). There were minor differences in the exact phrasing / chemical formula notation used.
> * For MPV, the agreement was over 80%, after we did a round of follow-up with annotators to adjudicate differences during a phase of pilot annotations.
> * For GEO, the agreement was again high after an initial pilot phase and discussion, agreement was over 90%, there were minor differences in exact phrasing e.g. {for each year between 2012-2015} vs {2012,2013,2014,2015}.

---

> ### Author Response · Authors · 2024-11-22
> **Response 2-of-2**
>
> **Question 1: LLMSim, nDCG**
> To clarify LLMSim a bit more, our retrieval tasks require exhaustive retrieval of all valid elements (e.g. materials), and each element being retrieved is a more complex dictionary, i.e. it is not just a material name but it also includes other parameters. Figure 10 and Figure 11  in the supplement provide an example each for DFT-S and MPV tasks. As noted earlier, the key evaluation challenge is identifying 1-to-1 mapping from the ground truth elements and model predictions, and each (key, value) in each dictionary needs to be verified for a match. LLMSim does a best effort 1-to-1 mapping of each ground truth element with a prediction, once the mapping is done we compute the precision and recall. The code for this is included in the supplement zip file.
>
> Thanks for suggesting alternate ideas like exact match and NDCG. We did try exact match but there are two main reasons why that fails, (i) material names and formulas can be written differently but could be equivalent, a simple case is illustrated in Figure 12 (HfO2 vs HFO<sub>2</sub>) other examples noted in Appendix C.2.4 include “Indium Nitride” vs “InN”, numeric values such as “100 $cm^2$” vs “1x$10^2 cm^2$” but there are many more complex cases. (ii) model responses do not parse into exact jsons, and not all keys/fields may be present in every dictionary element. So an exact match results in many more failures than necessary. And in this case ROUGE-L, which is recall based, already provides a reasonable substitute for exact-match.
>
> On the NDCG metric, NDCG is used to evaluate the quality of ranking of retrieved items with respect to an ordered list.  In our case there is no ordering for the elements, it’s a simpler retrieval task, the rank does not matter but the recall does. So the NDCG metric doesn’t apply to this task. Thanks for suggesting, and if you think there might be other
>
> **Question 2: Questions per paper**
> We have now clarified this in the table above. We have one question for each task per paper. DFT is one area where we have 3 tasks and each task has one question on each paper.  The HFD (derivation) task is also much more complex, but we treat all aspects of the prompt as a single question since that helps fully define the actual realworld derivation task.
>
> **Question 3: Avg. length**
> 954 is the average number of words in the ground truth responses. We have updated this figure in the revised draft.
>
> Thanks again for your positive comments and suggestions. We will work towards adding an open source evaluation for LMScore hopefully within this discussion period and adding additional questions to the dataset for some tasks as noted in our response.

---

> ### Comment · Area_Chair_7fxT · 2024-11-25
> **Reminder: Rebuttal Deadline for ICLR 2025**
>
> Dear Reviewer rvRN,
>
> As the rebuttal deadline approaches, please kindly check the papers' discussion threads and respond to the authors' rebuttals.
> If you haven't had a chance to respond yet, I’d greatly appreciate your input soon. Your insights are invaluable to the authors and the review process.
>
> Thank you for your effort and support!
>
> Best regards,
> Area chair

---

> > ### Author Response · Authors · 2024-11-26
> > **Request to acknowledge response**
> >
> > Dear reviewer rvRN,
> >
> > We have added additional details to the appendix based on your suggestions. In particular,
> >
> > * Numbers for the inter-annotator agreement for retrieval tasks.
> > * Appendix Tables 3 and 4, consolidating the tasks, number of examples, input and output lengths, brief description of tasks, assessed capabilities, and metrics.
> >
> > We also have an open source LLM based evaluation for the LMScore metric, that we are running, and are working on adding more examples for some task (BIOGR and PDB).
> >
> > We would appreciate it if you are able to acknowledge the response and let us know if these address your concerns to update your ratings or wish to share other comments.

---

> > > ### Comment · Reviewer_rvRN · 2024-11-26
> > > **Thank you for your responses!**
> > >
> > > Thanks for the detailed responses!
> > >
> > > I think most of my concerns have been addressed, so I’ve raised my score to 6. Personally, I would really like to use this dataset, so I hope the evaluation pipeline will be very easy to run. For example, it should be easy to test on a subset only and capable of generating an evaluation report for all datasets combined if evaluated on the entire dataset.

---

> > > > ### Author Response · Authors · 2024-11-27
> > > >
> > > > Thank you. Agreed! We hope to make it easy to run and evaluate on this benchmark. We have included notebooks to run inference and evaluations in the supplement, and also include code to generate the results tables and figures from the evaluations. We hope to make running and reporting of results straightforward.
> > > >
> > > > Thanks,

---

> > > > > ### Author Response · Authors · 2024-11-28
> > > > >
> > > > > A small update we have LMScore results from the open-source model as well for some of the tasks. We will include the evaluation code for these and results as well.
> > > > >
> > > > > |                         | GEO      | MPV      | HFD      | HFE      |
> > > > > |:------------------------|:---------|:---------|:---------|:---------|
> > > > > | command-r-plus          | 0.474022 | 0.550455 | 0.619123 | 0.369833 |
> > > > > | longllama               | 0.460661 | 0.590737 | 0.664614 | 0.293065 |
> > > > > | mixtral-gcp             | 0.545753 | 0.602009 | 0.571951 | 0.351843 |
> > > > > | gemini-1.5-flash-latest | 0.472547 | 0.728754 | 0.555545 | 0.393545 |
> > > > > | gemini-1.0-pro          | 0.44694  | 0.538038 | 0.512722 | 0.466863 |
> > > > > | gemini-1.5-pro-latest   | 0.479976 | 0.537297 | 0.575397 | 0.304802 |
> > > > > | gpt-4o                  | 0.486249 | 0.624556 | 0.625823 | 0.385702 |
> > > > > | claude-3-opus-20240229  | 0.452119 | 0.708169 | 0.578657 | 0.386113 |

---

### Author Response · Authors · 2024-11-23
**General response**

### We thank all reviewers for their positive comments as well as suggestions and feedback to help improve our work.

### All reviewers agree that a benchmark like CURIE for complex scientific domains is important (CbxB), helpful (zECm), a meaningful advance for AI4Science (EWc3), and would be beneficial for both developing better LLMs and better usage of LLMs for scientific research (dZb5). The reviewers also noted that the dataset is well-designed (rvRN) and carefully-collected (dZb5) with the help of domain experts (CbxB, dZb5), and the annotation process is thoroughly explained (rvRN) with the appendix including many details of the construction (dZb5). The reviewers also appreciated that the motivation for the sub-tasks were clearly articulated (rvRN) and the paper is well-written (CbxB, dZb5, zECm) and provides detailed experimental analysis and many meaningful discussions (CbxB).

Reviewers also included suggestions and requested clarifications. We include responses for some comments noted by multiple reviewers below.

**Number of examples, data size** (rvRN, CbxB)
We agree it would be good to have more examples for some tasks to provide stronger statistical signal. In fact, we did have more papers for GEO and BIOGR domains but had to remove them closer to submission when we discovered they had licenses that prevented re-distribution of the papers. Since submission, *we have gathered several more annotations for the BIOGR task, and are working to add more examples to the PDB tasks* as well to increase each to near 50 to provide a stronger statistical signal. We are still in the process of annotations and running evaluations. Just as a note of comparison, the now popular GPQA (https://arxiv.org/abs/2311.12022) dataset which tests expert level knowledge in biology, chemistry and physics contains 448 multiple choice questions; whereas our dataset covers disciplines requiring narrow expertise and has a wider range and complexity in responses which makes such a dataset more time and resource intensive to create. Thus, we believe the dataset and in particular annotations will have value despite the size.

**Evaluation of open-ended responses, and LLMSim** (rvRN, EWc3, CdxB)

We agree with reviewers that evaluation of open-ended responses is challenging. The advantage of ROUGE-L, BERTScore and LMScore metrics are that they do not require parsing of model output text and we agree these have limitations.

We note that for tasks where a structured output is preferred (e.g. GEO, DFT, MPV) our prompts provide the required JSON response format, and most LLM models – closed and open models – appear to adhere to instructions somewhat well to produce a JSON response in most cases. Our proposed LLMSim metric is used for some of these tasks with JSON responses. In the evaluation code included in the supplement zip file,  we incorporate several heuristics to give additional leeway when model responses do not conform to this format and do a best effort match between ground truth elements and model responses by making several calls to the LLM.

Overall, for the retrieval tasks such as DFT-S, DFT-P, MPV, we recommend LLMSim since it is designed and better suited for these retrieval tasks. For summarization and aggregation tasks such as HFE, HFD, QECC, and DFT-C, the existing ROUGE and BERTScore metrics are appropriate as is. We use LMScore as an additional model based evaluation to provide signal, and is not necessarily meant to replace other metrics just yet. For BIOGR, we  use Intersection Over Union; and for PDB we use ID_{ratio} which are both designed by domain experts and appropriate metrics well suited for the tasks.


**Automatic scaling of dataset** (zECm, dZb5)

We agree such dataset creation requires human (and actually expert) annotations and scaling this is a challenge. CURIE is a new benchmark that fills in the gap in existing literature and curate a range of problem solving tasks in science. For instance, early works such as [1], which aimed to look at the impact of LLMs for science were a collection of anecdotal examples noted by experts. In contrast, our work aims to provide a concrete measurable benchmark. Similarly, the very recent FrontierMath [2] benchmark was also created with the help of expert mathematicians and professors coming together, and required human annotation. As we progress, we believe we might be able to identify datasets and tasks from experts in different scientific fields that might enable easier scaling. For instance, the PDB task in our dataset is quite amenable to scaling since there is a programmatic way to generate the solutions and the evaluation is also much more concrete. With newer agentic frameworks developing, we will think more about how examples may be automatically updated and verified for some of these tasks. We can include this as a future direction.


[1] https://arxiv.org/abs/2311.07361

[2] FrontierMath: https://arxiv.org/abs/2411.04872

---

> ### Author Response · Authors · 2024-11-23
> **General response -- Draft edits**
>
> Based on the nice suggestions and feedback from the reviewers, we hope to include some edits in the draft during the discussion phase and some more soon afterwards. We will make the following edits to the draft:
>
> 1.  Include a summary table with number of examples, papers, input and output lengths in a consolidated table format in the appendix. (included in response to rvRN)
> 2. Include numbers for inter-annotator agreement. (included in response to rvRN)
> 3. Include prompts for most tasks in the appendix. (EWc3, CbxB)
> 4. Include a figure that will help correlate performance on CURIE with respect to other Science QA tasks and Long-context benchmarks. (CbxB)
>
> We are also working on adding these but likely after the discussion period:
>
> 5.  Add more examples for the BIOGR and PDB tasks to provide greater statistical strength. (rvRN, CbxB)
> 6.  Add an open-source LLM based version of LMScore evaluation. (rvRN)
>
> We will attempt to look into creating a multiple choice version of the GEO task if possible (CbxB)
>
> We would like to thank the reviewers again for their time and thoughtful comments and look forward to addressing any other questions during this discussion phase.

---

> ### Author Response · Authors · 2024-11-23
> **Updated PDF draft with reviewer suggestions**
>
> We have updated the draft to include the following for suggestions from the reviewers:
>
> 1. A summary table with number of examples, papers, input and output lengths in a consolidated table format (included in response to rvRN)  -- is now in Appendix A
> 2. Statistics for inter-annotator agreement. (included in response to rvRN) -- is now in Appendix C
> 3. Included prompts for most tasks in the appendix. (EWc3, CbxB) -- Added 10 figures, one for each task in the appropriate appendix subsections.
> 4. Included a figure that will help correlate performance on CURIE with respect to other Science QA tasks and Long-context benchmarks. (CbxB) -- is now in Appendix B

---

> > ### Author Response · Authors · 2024-11-26
> > **Additional updates to the PDF with reviewer suggestions**
> >
> > 5. We have added a table to consolidate the tasks, capabilities evaluated, the output format, and metrics for evaluation. (included in response to EWc3)
> >
> > In progress:
> >
> > We have implemented open source model version of LMScore on the Gemma 2B model that can be easily run in a free colab and are running evaluations (rvRN). We hope to add these soon.
> >
> > We are also gathering more BIOGR and PDB examples for the benchmark.

---

### Meta-Review · Area_Chair_7fxT · 2024-12-19

**Metareview:**

Summary of the paper:
This paper introduces CURIE, a benchmark designed to evaluate LLMs on scientific problem-solving tasks that require long-context understanding and multi-step reasoning across six disciplines: materials science, condensed matter physics, quantum computing, geospatial analysis, biodiversity, and proteins. CURIE consists of ten challenging tasks derived from 434 examples taken from 273 research papers, emphasizing realistic scientific workflows such as information extraction and concept tracking. The authors assess various LLMs, noting that Claude3 consistently outperforms others, including GPT4, in specific tasks like protein sequencing. The study identifies significant performance gaps across LLMs, highlighting the need for improvements in LLM capabilities for scientific applications. Additionally, a new LLM-based evaluation metric is introduced, along with evaluations conducted by domain experts on LLM predictions.

Strengths of the paper:
- Comprehensive Benchmark: Introduces the CURIE benchmark with 434 examples from 273 science literature, validated by domain experts. Covers ten complex tasks across six scientific fields, targeting realistic problems faced by scientists.
- Contribution to Scientific Research: Provides a valuable tool for the development and application of language models in scientific contexts.
- Detailed Experimental Analysis: Assesses various LLMs and discusses challenges faced by LLMs in long-context reasoning, highlighting their limitations and areas for improvement in scientific reasoning tasks.

Weaknesses of the paper:
- Small Dataset Size (Reviewers rvRN, CbxB): CURIE consists of only 434 examples, with certain tasks like GEO and BIOGR having as few as 19-21 examples, limiting statistical significance and reliability at the sub-task level.
- Challenges in Dataset Scaling (Reviewers zECm, dZb5): The requirement for human annotations makes it difficult to scale the dataset effectively, potentially hindering future research. Additionally, concerns exist about whether LLMs may have been trained on the same research papers included in the CURIE benchmark, affecting evaluation outcomes.
- Evaluation Difficulties with Open-ended Tasks (Reviewers rvRN, EWc3, CdxB): For open-ended tasks, the variability in model responses makes it challenging to assess correctness, particularly when using different notations or steps. The evaluation relies on GPT-4 to compute LMScore, which may vary over time due to updates. Additionally, the correlation between LMScore and human judgment is not convincingly established, raising doubts about its effectiveness compared to ROUGE.
- Lack of Clarity in Presentation (Reviewers rvRN, EWc3): The paper could use tables to make the presentation clearer.

Reasons for the decision:
After considering the rebuttal, I believe that most of the concerns raised have been adequately addressed by the authors, as outlined in the reviewer discussion box. All reviewers agree that this paper should be accepted at ICLR. Upon careful reflection, I find this paper to be a valuable resource, as it delves into an unexplored area within AI for science. It has the potential to engage the broader community and make an impact. However, I recommend that the authors release updated benchmarks with additional annotations for the BIOGR and PDB tasks for the camera-ready version of the paper.

**Additional Comments On Reviewer Discussion:**

Thanks to the authors for summarizing the comments during the rebuttal.

- Small Dataset Size (Reviewers rvRN, CbxB): The authors claimed that they have gathered several more annotations for the BIOGR task, and are working to add more examples to the PDB tasks as well to increase each to nearly 50 to provide a stronger statistical signal. On the other hand, the now popular GPQA dataset only contains 448 multiple-choice questions; whereas CURIE covers disciplines requiring narrow expertise and has a wider range and complexity in responses.
- Evaluation Difficulties with Open-ended Tasks (Reviewers rvRN, EWc3, CdxB): The authors clarify that for the retrieval tasks such as DFT-S, DFT-P, MPV, we recommend LLMSim since it is designed and better suited for these retrieval tasks. For summarization and aggregation tasks such as HFE, HFD, QECC, and DFT-C, the existing ROUGE and BERTScore metrics are appropriate as is. We use LMScore as an additional model based evaluation to provide signal, and is not necessarily meant to replace other metrics just yet. For BIOGR, we use Intersection Over Union; and for PDB we use ID_{ratio} which are both designed by domain experts and appropriate metrics well suited for the tasks.
- Lack of Clarity in Presentation (Reviewers rvRN, EWc3): The authors add the tables and update the PDF draft based on reviewers' suggestions.

---

### Decision · Program_Chairs · 2025-01-22

Accept (Poster)